# Underestimated barrier effects of ocean fronts shape global fishery distribution

Qinwang Xing [1,2,3,4,5] ✉, Zihui Gao[1,2], Shin-ichi Ito [2], Haiqing Yu [6], Bilin Liu[1,3,4,5], Heng Zhang [7], Xinjun Chen [1,3,4,5] ✉ & Wei Yu [1,3,4,5] ✉

Marine ecosystems exhibit high spatiotemporal heterogeneity, making it crucial to understand the mechanisms sustaining biological hotspots. Ocean fronts shape local biogeochemical processes and have long been recognized as biological hotspots aggregating organisms from phytoplankton to top predators and attracting fisheries (hotspot effects). However, fronts also exhibit pronounced environmental differences between their two sides (barrier effects), and how species and fishery distributions respond to these effects remains poorly understood. By integrating satellite-based front detection with commercial catch records, fishery-independent surveys, and global fishing datasets, we show that fishery distributions across diverse regions and major commercial stocks worldwide respond strongly to barrier effects, exhibiting 15–70% differences in distribution between the frontal warm and cold zones, driven by species-specific local thermal preferences. In contrast, responses to hotspot effects are generally sporadic with only 5–20% differences between frontal and non-frontal zones, and they emerge only when aggregations on one side of fronts offset avoidance on the other. This offset has led earlier studies to conservatively underestimate front-induced fishery variations by 55–75%. Our findings complement the traditional front-induced hotspot paradigm by clarifying the importance of barrier effects and underscore the need to reassess the role of ocean fronts in marine ecosystems.

Marine organism-environment interactions operate across multiple spatial and temporal scales. Globally, marine productivity peaks in mid-latitude oceans and declines to oligotrophic deserts within tropical and subtropical gyres, driven by light and nutrient gradients[1]. At the basin scale, energetic circulation and upwelling enhance nutrient input to the euphotic zone, generating high-productivity regions along western boundary current extensions and eastern boundary upwelling systems[2]. These areas sustain elevated biodiversity and support some of the world's most productive fisheries[3], such as the Kuroshio-Oyashio extension (KOE)[4]. At regional scales, the distribution patterns of marine organisms are highly variable and are increasingly being attributed to mesoscale oceanographic features spanning tens to hundreds of kilometers[5,6]. Through intense physical-biological-biogeochemical interactions, these features are thought to generate biological hotspots ranging from plankton to top predators, and in turn attract frequent human predation via fishing[7–9]. Such features form a critical foundation for fine-scale distribution prediction and modern dynamic ocean management[10,11], and have therefore received growing attention

[1]College of Marine Living Resource Sciences and Management, Shanghai Ocean University, Shanghai, China. [2]Atmosphere and Ocean Research Institute, The University of Tokyo, Kashiwa, Japan. [3]National Distant-water Fisheries Engineering Research Center, Shanghai, China. [4]Key Laboratory of Sustainable Exploitation of Oceanic Fisheries Resources (Shanghai Ocean University), Ministry of Education, Shanghai, China. [5]Key Laboratory of Sustainable Utilization of Oceanic Fisheries, Ministry of Agriculture and Rural Affairs, Shanghai, China. [6]Institute of Marine Science and Technology, Shandong University, Qingdao, China. [7]East China Sea Fisheries Research Institute, Chinese Academy of Fisheries Science, Shanghai, China. ✉e-mail: qwxing@shou.edu.cn; xjchen@shou.edu.cn; wyu@shou.edu.cn

to understand how organisms exploit mesoscale features, particularly for fish species and fisheries with high ecological and economic value[8,9,12].

Ocean fronts are discrete yet ubiquitous mesoscale dynamic features[13], typically defined as narrow, high-energy boundary zones where water masses with contrasting temperature, salinity, and other properties converge[14]. Front-induced secondary circulation and enhanced turbulent mixing substantially increase nutrient flux into the sunlit surface layer, sustaining prolonged phytoplankton blooms[6,15,16]. Frontal convergence effects also lead to the aggregation of drifting phytoplankton and zooplankton, forming high-quality spawning, foraging, and migratory habitats for higher trophic-level species, including fishes, seabirds, and marine mammals[9,17–19]. The close association between marine animals and fronts was first documented as early as the 17th century[20], and fishers have long used fronts as indicators of productive fishing grounds to locate and track fish schools[9,12,21]. However, it was not until the late 20th century—driven by advances in satellite observations and automated front detection algorithms[22–24]—that statistical analyses integrating frontal distribution maps derived from satellite-based sea surface temperature (SST) imagery with fishery distribution and GPS-tracked animal movements provided compelling evidence that fishes, top predators, and fishing activities tend to aggregate along frontal zones, largely due to enhanced foraging opportunities arising from predator-prey interactions[9,12,21,22].

This classical paradigm has been extensively applied over the past three decades to the spatiotemporal analysis and prediction of fish and marine animal habitats for conservation and management purposes[9,19,25,26]. However, recent case studies suggest that the distributions of some fishery species, fishing activities, and top predators are not consistently linked to ocean fronts, which deviates from the prevailing paradigm that fronts serve as biological hotspots[17,27–29]. For example, statistical analyses of trolling and bait fleet data in the northeast Atlantic revealed that catches of albacore tuna (see Table S1 for scientific names) lacked the expected strong association with fronts, contrasting earlier findings based on visual interpretation[29]. Similar cases were also observed for salmon and pink shrimp fisheries in the California upwelling region[28], seabirds and marine mammals in New Zealand waters[27], and blue and hammerhead sharks in the North Atlantic[17,30], although other species in these studies do show strong aggregations at fronts. Additionally, some previous studies reported that frontal proxies (frequency, gradient, and distance) generally contribute little to statistical models, indicating an unexpectedly limited effect of fronts on biological distributions[26]. The mechanisms underlying these inconsistent responses relative to the prevailing theory remain poorly understood, highlighting the need to re-evaluate the ecological and fisheries significance of ocean fronts to clarify these observed deviations.

Temperature is widely recognized as a key driver of species distributions, particularly for ectotherms such as most marine species[31]. Deviations from thermal optima reduce aerobic scope, impair growth and predation efficiency, and ultimately suppress population abundance[32]. Thus, in addition to serving as biological hotspots, ocean fronts act as thermal boundaries that can exert strong barrier effects on marine organisms[33]. For example, field surveys have shown that zooplankton are more abundant on the colder, shelf side of the Georges Bank front and less abundant on the warmer, slope side[34], while juvenile rockfishes and jack mackerel are restricted offshore by coastal upwelling fronts[35]. However, a comprehensive assessment that simultaneously considers both hotspot and barrier effects remains lacking[26], leaving the responses of fish species with different thermal preferences and their associated fishery distributions to frontal warm and cold zones poorly understood. A major challenge lies in reliably detecting ocean fronts from satellite-derived sea surface temperature data[9], where most existing detection algorithms were designed to locate frontal positions for studying frontal hotspot effects, rather than to distinguish between the warm and cold sides within frontal zones[23,24,36].

In this study, we apply a recently developed satellite-based front detection method that overcomes known limitations of earlier approaches and enables the automated identification of both warm and cold zones on either side of each front[37,38], and we establish an analytical framework to quantify frontal hotspot and barrier effects on fisheries. By integrating this approach with commercial catch data, fishery-independent survey data, and deep learning-derived fishing activity data for key stocks in the KOE, we reveal that fisheries respond strongly to frontal barrier effects, generally tending to aggregate in either the frontal warm or cold zones while avoiding the other, primarily driven by species-specific thermal preferences at local and intra-seasonal scales. Extending our analysis to globally representative regions and stocks using industrial fishery big data[39], we find that frontal barrier effects are widespread, whereas hotspot effects are less universal than previously assumed, emerging only when aggregations on one side of fronts offset avoidance on the other. These responses to barrier effects differ from earlier views, which primarily emphasized frontal barriers as limiting material transport, such as plankton, larvae, and gene exchange[12], and separating species distributions across large-scale biogeographical boundaries, such as the subarctic or polar fronts[40,41]. Instead, our findings reveal that different fishery-targeted species actively share and exploit habitats structured by highly dynamic and ubiquitous mesoscale frontal systems through species- and season-specific thermal strategies.

## Results
### Species-specific responses to frontal hotspot effects
The convergence of the Kuroshio and Oyashio currents in the KOE region creates one of the world's most dynamic frontal systems, characterized by exceptionally high biological productivity[42,43]. With frequent and intense frontal activity and harboring diverse target species with contrasting thermal affinities, the KOE provides a natural laboratory for disentangling the effects of mesoscale fronts on fisheries[44]. We compiled >100,000 daily fishing records from commercial fishing logbooks targeting four of the region's most productive and economically important stocks (Fig. 1a), spanning distinct thermal ecological niches, including warm-affinity neon flying squid, cold-affinity Pacific saury, and warm eurythermal chub mackerel and Japanese sardine[31,45]. To quantify the frontal hotspot effects, we integrated these fishery records with daily satellite-detected frontal distributions and compared catches between frontal and non-frontal zones using two independent statistical indices: the relative difference in fishing catch (or effort) per unit area ($RD_{FPA}$)[7], and the fishing catch (or effort) relative anomaly difference (FRAD), defined as the difference in the relative deviation of observed total catches (or efforts) relative to the large-scale fishery background distribution[46]. Statistical significance was assessed using a bootstrap method (see Methods section).

FRAD results indicate that mackerel and sardine catches increase by ~50% within frontal zones compared to non-frontal zones, while saury shows a slight increase by ~17% (Fig. 1b). Contrary to expectations, squid catches decline slightly by ~13% within frontal zones. These patterns are corroborated by $RD_{FPA}$ results, and t-tests reveal significant differences (all $p < 0.01$, see Source Data for full statistics, hereafter) between the observed and 1000 simulated values for all four species, supporting the robustness of the findings (Fig. 1b). The spatial distribution of $RD_{FPA}$ values demonstrates strong regional and seasonal consistency for sardine and mackerel, whose catches increase consistently and significantly within frontal zones, whereas saury and squid exhibit greater spatial variability and uncertainty (Fig. 1c–f). Since our commercial fishing records cover only part of the fishing

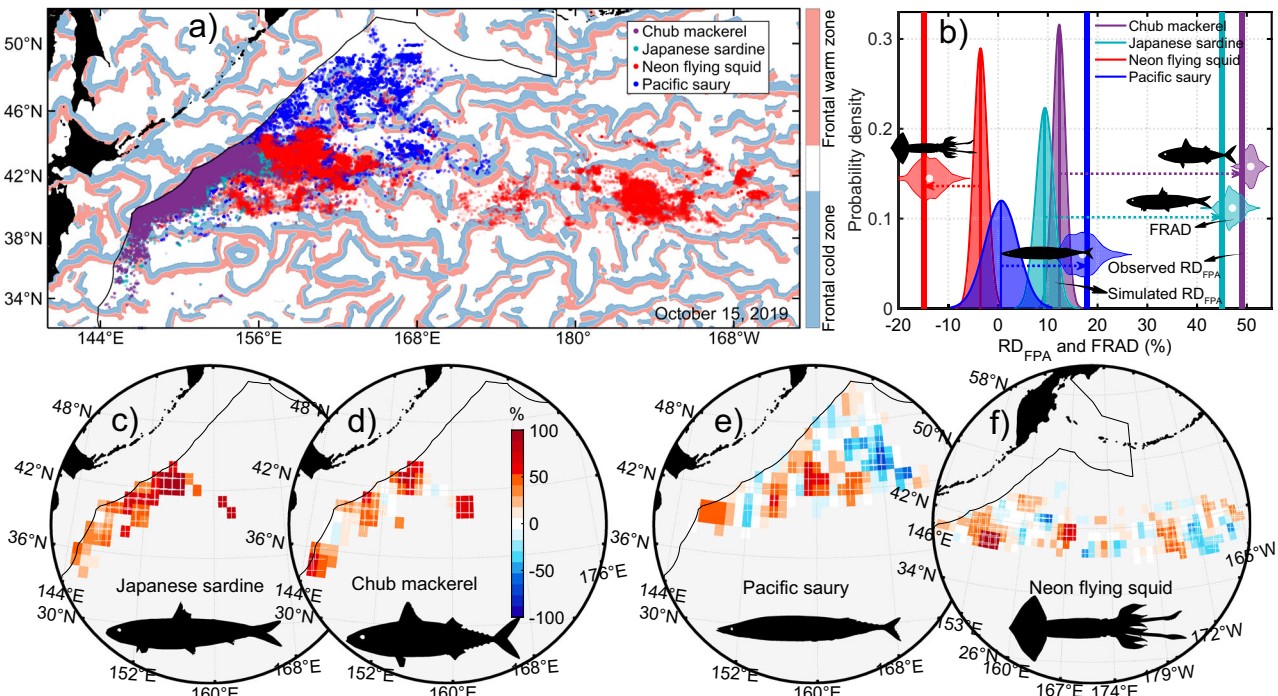

**Fig. 1 | Frontal biological hotspot effects on fisheries in the Kuroshio–Oyashio extension (KOE). a** Satellite-detected frontal warm and cold zones on 15 October 2019, overlaid with the distribution of all commercial fishing records for the four species analyzed in this study. Fishing locations were randomized to protect fishery data privacy. **b** Fishery responses to frontal hotspot effects, quantified by differences in catch between frontal and non-frontal zones. Vertical lines indicate the observed relative difference in fishing catch per unit area ($RD_{FPA}$), overlaid on the probability density distributions of $RD_{FPA}$ generated by random simulations (bell-shaped areas). Horizontal violin plots show the fishing catch relative anomaly difference (FRAD). Positive (negative) values of $RD_{FPA}$ and FRAD indicate higher (lower) catches in frontal zones compared to non-frontal zones. **c–f** Spatial distribution of $RD_{FPA}$ between frontal and non-frontal zones for the four species. White plus signs indicate areas with statistically significant $RD_{FPA}$, and grey areas denote regions with no data. Red areas indicate higher catches in frontal zones relative to non-frontal zones, while blue areas indicate the opposite. The figure shows strong positive responses of mackerel and sardine fisheries, a weak response of saury, and a negative response of squid to frontal hotspot effects. Source data are provided as a Source Data file.

grounds and do not include fisheries within exclusive economic zones, we conducted a parallel analysis using deep learning-derived global industrial fishing activity[39]. This independent dataset reveals a similar spatial pattern and provides additional support for our findings (Supplementary Fig. S1). Overall, results from diverse datasets and analytical methods consistently show that front-induced hotspot effects on fishery distribution are species-specific rather than universal: mackerel and sardine fisheries respond strongly and positively, saury shows a weak response, and squid respond negatively. This provides direct and quantitative evidence supporting recent departures from the widely accepted theory of front-induced hotspots. It should be noted that catch differences from FRAD and $RD_{FPA}$ are not direct quantifications; rather, they eliminate biases resulting from the uneven areas occupied by frontal (including their warm and cold zones) and non-frontal zones.

### Frontal barrier effects explain species-specific hotspot effects

To clarify species-specific patterns of frontal hotspot effects and the influence of barrier effects on fishery distributions, we compared catches between frontal warm and cold zones using FRAD and $RD_{FPA}$. FRAD and $RD_{FPA}$ results consistently show that mackerel and sardine catches exhibit minimal differences (5–15%) between frontal warm and cold zones, but increase substantially (by 45–65%) in both zones relative to non-frontal waters (Fig. 2a, b). In contrast, saury and squid catches exhibit pronounced differences of ~40% and ~70%, respectively: saury catches increase by ~35% in frontal cold zones but decrease by ~6% in warm zones, whereas squid catches rise by ~20% in warm zones and decline sharply by ~45% in cold zones (Fig. 2a, b). Based on the 1000 simulated values, t-tests confirm the statistical

significance of these differences for saury and squid (all $p < 0.01$). Spatial patterns in $RD_{FPA}$ reveal consistent responses of saury and squid catches to frontal barrier effects across regions and seasons (Fig. 2d, e). In contrast, mackerel and sardine display opposing responses between northern and southern parts of the fishing grounds (Fig. 2f, g), with catches increasing within frontal cold zones and decreasing within warm zones in the south, but showing the opposite pattern in the north. These opposing trends result in weak region-integrated responses of mackerel and sardine to frontal barrier effects (Fig. 2a).

These species undertake seasonal migrations between the Kuroshio and Oyashio waters, driving north-south shifts in their fishing grounds[45]. To assess seasonal variability in frontal barrier effects, we examined catch relative anomalies as a function of normalized distance from fronts for each fishing month. The responses of mackerel and sardine catches to barrier effects reverse before and after August (Fig. 2h–k), thereby explaining their north-south spatial shifts. Squid catches exhibit consistent responses throughout the fishing season, with higher catches in warm zones and lower in cold zones, whereas saury responds strongly only after August, preferring cold zones and avoiding warm zones, with negligible response in June and July. Meanwhile, these patterns are further supported by a generalized additive model and independently validated using deep learning-derived fishing activity data, providing an additional method and dataset (Supplementary Figs. S2–S3). Furthermore, we conducted fishery-independent surveys from June to July during 2021–2024. Although saury was not captured in these surveys, the results still show a significantly higher CPUE in frontal warm zones than in cold zones for mackerel, sardine, and squid (all p < 0.05, Wilcoxon signed-rank test),

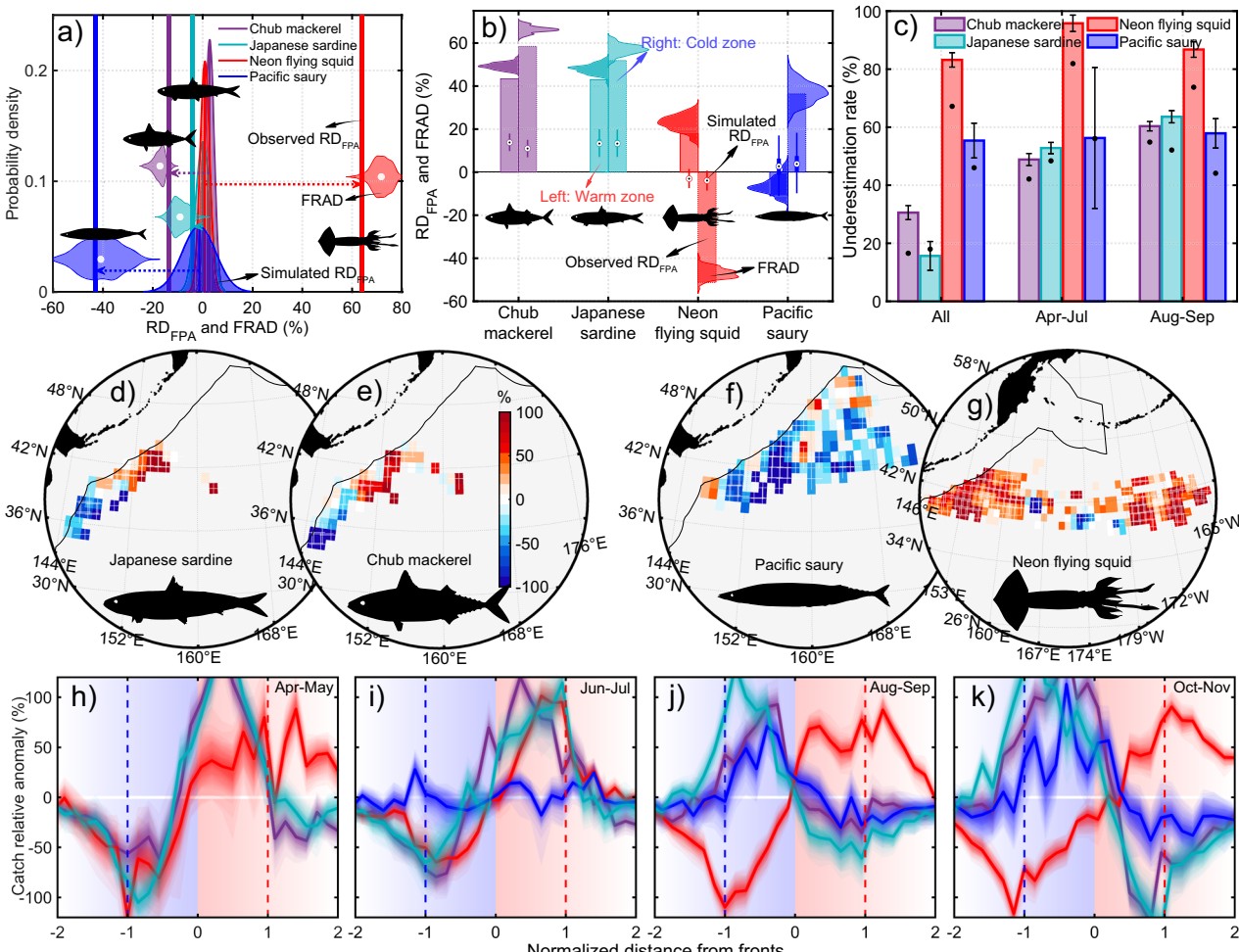

**Fig. 2 | Frontal barrier effects on fisheries in the Kuroshio–Oyashio extension (KOE). a** Similar to Fig. 1b, but showing fishery responses to barrier effects, quantified by differences in catches between frontal warm and cold zones. Positive (negative) values indicate higher (lower) catches in warm zones compared to cold zones. **b** Catch differences between frontal warm (left) or cold (right) zones and non-frontal zones. Horizontal bell-shaped areas and bar graphs show the fishing catch relative anomaly difference (FRAD) and observed relative difference in fishing catch per unit area ($RD_{FPA}$), respectively. Boxplots show the non-outlier minimum/maximum and the 25th, 50th, and 75th percentiles of 1000 randomly simulated $RD_{FPA}$. **c** Underestimation rate of front-induced variations in fishery distribution over fishing seasons when barrier effects are ignored, calculated as the percentage difference between the absolute values of hotspot and barrier effects relative to the larger of the two using FRAD. Error bars indicate standard deviations based on 1000 simulations, with the central value representing the mean. Black points show underestimation rates from $RD_{FPA}$. **d–g** Similar to Fig. 1c–f, but showing $RD_{FPA}$ between frontal warm and cold zones. Red areas indicate higher catches in warm zones relative to cold zones, while blue areas indicate the opposite. **h–k** Catch relative anomaly at different normalized distances from fronts for each fishing season. Positive (negative) distances correspond to the warm (cold) side of fronts; red (blue) dotted lines mark the normalized boundaries of warm (cold) zones. Lines show medians from 1000 random simulations; shading represents different percentile ranges. The figure shows strong positive and negative responses of squid and saury fisheries, respectively, and seasonally reversed responses (before and after August) of mackerel and sardine to frontal barrier effects. Source data are provided as a Source Data file.

providing fishery-independent support for our observed pattern before August (Supplementary Fig. S3e).

Overall, results from diverse datasets and analytical methods consistently show that mesoscale fronts exert significant barrier effects on the distributions of all four fishery targets, with effects that are species-specific and seasonally variable. Interestingly, the consistently positive responses of mackerel and sardine fisheries to both warm and cold frontal zones align with their strong positive association with frontal hotspot effects. In contrast, squid catches show a weak negative association with frontal hotspot effects, as strong avoidance of cold zones offsets mild preference for warm zones; saury shows a weak positive association, driven by strong affinity for cold zones and slight avoidance of warm zones. For the four major species in the KOE, ignoring frontal barrier effects leads to a substantial underestimation of frontal impacts (Fig. 2c). Relative to the maximum differences between frontal warm and cold zones, FRAD

results show that region-integrated hotspot effects are underestimated by 30%, 15%, 80%, and 55% for mackerel, sardine, squid, and saury catches, respectively. The underestimation for mackerel and sardine increases to 50–60% when fishing months before and after August are analyzed separately, whereas estimates for squid and saury show only minor seasonal variation (Fig. 2c). These underestimations are further supported by $RD_{FPA}$ results. These frontal barrier effects can explain the species-specific nature of frontal hotspot effects (Fig. 1) and suggest that deviations from the traditional frontal hotspot paradigm may stem from overlooking frontal barrier effects[27,28].

## Seasonal local thermal preferences regulate frontal barrier effects

To clarify species- and season-specific responses of fishery distributions to frontal barrier effects, we hypothesized that front-related

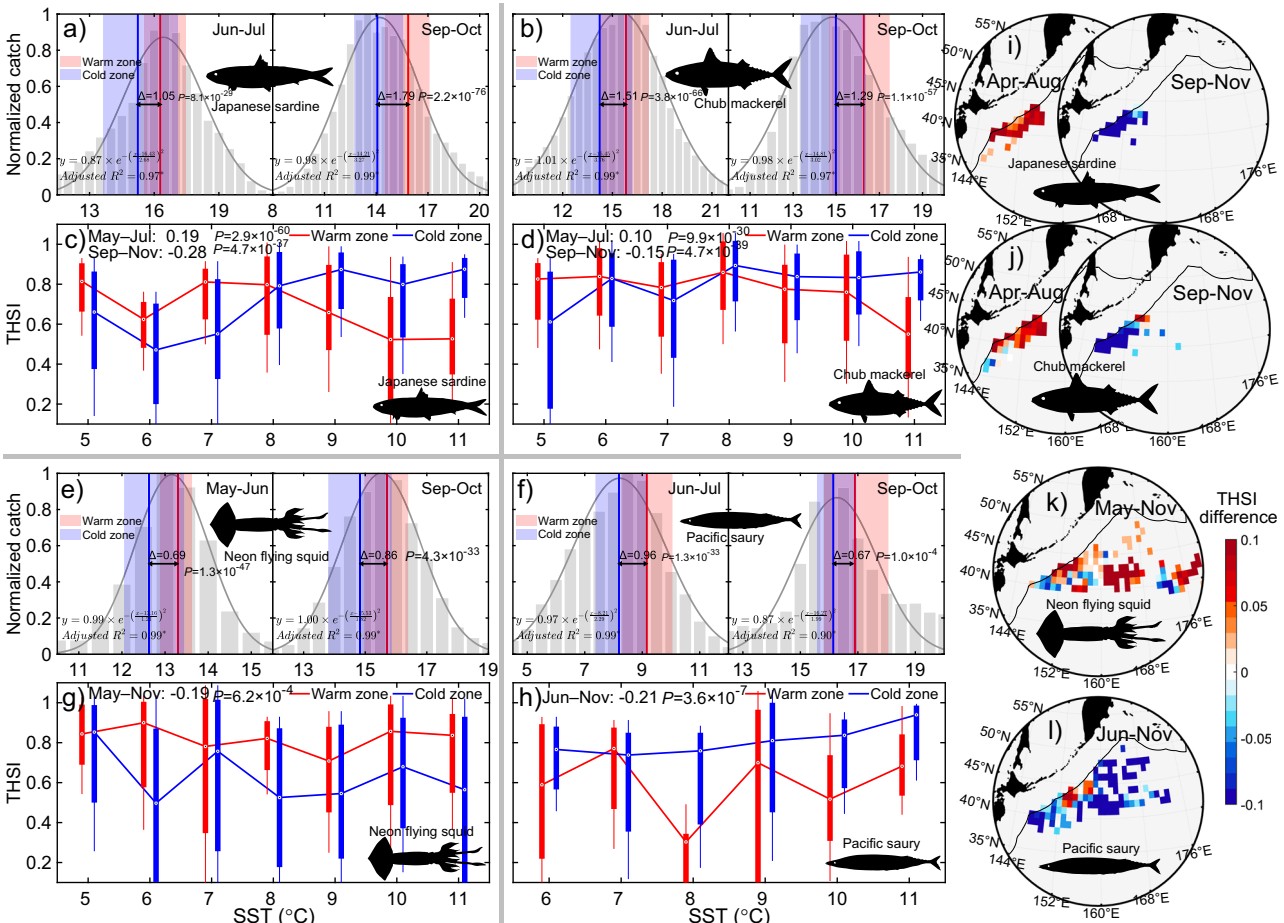

**Fig. 3 | Front-related thermal preferences and habitat dynamics of the four species. a–b, e–f** Normalized catches across SST bins (grey bars) and their Gaussian fits (grey lines) for the four species before and after August. Vertical red and blue lines along the x-axis indicate the median SSTs of all highly productive fishing locations (those in the top 50% of catch per unit effort) within frontal warm and cold zones, respectively, with shaded areas representing the 25th and 75th percentiles. *P*-values indicate the significance of their SST differences between warm and cold zones based on two-sided t-tests. **c–d, g–h** Predicted thermal habitat suitability index (THSI) in highly productive fishing locations across months. Red and blue boxplots show THSI values in warm and cold zones, respectively, indicating the non-outlier minimum/maximum and the 25th, 50th, and 75th percentiles. *P*-values indicate the significance of THSI differences between zones based on two-sided t-tests. All exact statistics are available in the Source Data. **i–l** Spatial distribution of THSI differences between warm and cold zones at highly productive fishing locations. Red (blue) areas indicate higher (lower) thermal habitat suitability in warm zones relative to cold zones, while grey areas denote regions with no data. The figure shows that squid and saury exhibit higher thermal habitat suitability in warm and cold zones, respectively, while mackerel and sardine show a seasonal shift in thermal preference, preferring warm zones before August and cold zones after August. Source data are provided as a Source Data file.

variations in catches and fishing activity primarily reflect shifts in fish distribution driven by a bottom–up response to prey movement[9,12]. Given that marine species are sensitive to local environmental variability driven by mesoscale ocean dynamics[5,26], we first compared the environmental conditions between frontal warm and cold zones using independent subsurface and satellite-derived surface observations (Supplementary Fig. S4). Compared to frontal cold zones, consistently higher subsurface (200 m) temperatures in warm zones across regions and seasons support the robustness of our analytical framework. In addition, warm zones consistently exhibit lower levels of dissolved oxygen and chlorophyll (a proxy for food availability) than cold zones both before and after August within these fishing grounds. These findings suggest that the observed fishery responses to frontal barrier effects cannot be attributed to front-induced environmental differences, particularly for the season-specific reversals.

Given that species typically occupy distinct thermal niches, we next assessed thermal preferences and habitat dynamics using a niche-based thermal habitat suitability model (see Methods section). All four species exhibit distinct seasonal shifts in thermal preferences before and after August, with SSTs at highly productive fishing locations (top 50% CPUE, catch per unit effort) in frontal warm and cold zones differing significantly (all *p* < 0.01) between seasons (Fig. 3a, b, e, f). Notably, before August, SSTs at highly productive fishing locations in frontal warm zones align more closely with the preferred temperatures of mackerel and sardine than SSTs in cold zones, whereas after August, cold zones better match their thermal preferences. In contrast, SSTs in frontal warm zones consistently align with the thermal preferences of squid, while cold zones better match those of saury throughout the season. Predicted thermal habitat suitability indices at high-CPUE sites differ significantly (all *p* < 0.01) between warm and cold zones for all four species (Fig. 3), reflecting seasonal and spatial patterns consistent with observed frontal barrier effects on fisheries (Fig. 2). For mackerel and sardine, thermal habitat is significantly higher in warm zones with strong spatial consistency before August and this pattern reverses afterward, whereas squid and saury show higher habitat suitability in warm and cold zones across all seasons, respectively. Overall, the thermal ecological niche hypothesis indicates that variations in frontal barrier effects across fisheries are largely explained by species- and season-specific thermal preferences in local waters.

## Frontal fishery impacts widely underestimated due to barrier effects

To assess whether our findings reflect a local phenomenon or a broader global pattern, we analyzed a deep learning-derived global fishing activity dataset to examine frontal biological hotspot and barrier effects on fishery distributions across 11 representative regions dominated by major boundary currents and upwelling systems[43], spanning 6 major gear types. Results from 1000 random simulations indicate that 93.0% of fisheries (53 out of 57) exhibit significant ($p < 0.05$) frontal barrier effects on fishing activities, and 75.4% (43 out of 57) show significant hotspot effects (Supplementary Fig. S5a–k). Based on FRAD, pole-and-line, other purse seines (targeting non-tuna species), and trawlers respond most strongly to barrier effects, with average fishing activity differences between frontal warm and cold zones reaching 40–70%, while drifting longlines show a weaker response (-15%) (Fig. 4a). The weak response of drifting longline fisheries to barrier effects may stem from their long-range operations (50–100 km)[39], which can decouple recorded fishing locations from actual catch sites and thus dilute finer-scale front-fishery associations. Across all gear types, responses to frontal hotspot effects are significantly weaker than responses to barrier effects ($p < 0.01$, Mann-Whitney U test), with other purse seines and trawlers showing the highest differences (20–25%) between frontal and non-frontal zones, while drifting longlines, pole-and-line, and tuna purse seines show differences of only 5–10%. These findings are independently supported by the $RD_{FPA}$ results (Supplementary Fig. S6).

Given the ambiguity in target species associated with these fishing gears and the potential confounding effects of multispecies aggregation at fronts, we further evaluated 25 fisheries targeting the most commercially important and globally recognized stocks (see Methods section). FRAD results show a difference of 36% ± 15% in fishing activity between warm and cold zones, significantly greater ($p < 0.01$, t-test) than 12% ± 13% between frontal and non-frontal zones (Fig. 4c). Similarly, $RD_{FPA}$ results show differences of 33% ± 15% and 11% ± 13%, respectively (Supplementary Fig. S6). Peruvian anchovy—the world's most productive stock—and Antarctic krill—the largest biomass stock globally—show the strongest negative responses to frontal barrier effects, with 55–65% more fishing activity in cold zones. Alaska pollock (the second most productive fishery), skipjack tuna in the Northwest Pacific, Atlantic cod in the Barents Sea, albacore tuna and swordfish in the Southwest Atlantic, and troll fisheries targeting albacore tuna across all oceans exhibit the strongest warm-zone preference, with 40–70% higher fishing activity than in cold zones. In response to barrier effects, most stocks show a strong preference for either the warm or the cold side of fronts, while exhibiting an opposite or weaker response on the other side (Supplementary Fig. S6d). Meanwhile, simulated results show that fisheries for nearly all assessed stocks respond significantly to barrier effects ($p < 0.05$), except for Japanese sardine and chub mackerel, which are primarily influenced by seasonal variations (Fig. 2). In contrast, over half (13 out of 25) exhibit no significant ($p > 0.05$) response to hotspot effects, and four even exhibit significantly negative responses, with 8–29% lower fishing activity in frontal zones than in non-frontal waters. These negative responses reflect asymmetrical influences, where weaker benefits from one side of frontal zones fail to offset adverse effects from the other (Supplementary Fig. S6d).

These results challenge the widely accepted paradigm of front-induced biological hotspots and underscore the species-specific responses to fronts. Critically, frontal effects on fishery distributions are significantly underestimated across 91% of regional cases and in all 25 fishery stocks ($p < 0.05$, t-test), with mean underestimation rates of 64% ± 22% and 75% ± 18%, respectively (Fig. 4b, c and Supplementary Fig. S5l). None of the regions or stocks without underestimation reaches statistical significance in our analysis. Independently, $RD_{FPA}$ results support these findings, showing mean underestimation rates of 56% ± 23% and 67% ± 24%, respectively. It is important to note that these underestimation estimates are conservative, as potential seasonal reversals in responses to frontal barrier effects, such as those observed for sardine and chub mackerel in the KOE (Fig. 2c), may not be fully captured. In addition, deep learning-derived fishing effort data may include gear type misclassifications and inaccuracies in fishing hours[39], and together with errors from satellite-based front detection[38], these sources of noise can dampen the detected strength of frontal effects and further lower the estimated underestimation rate. Overall, these results provide strong evidence that frontal barrier effects on fisheries are widespread across the global ocean and ignoring them results in a substantial 55–75% underestimation of frontal fishery impacts.

## Discussion

In this study, we developed a framework for analyzing front-induced fishery effects by integrating an advanced mesoscale front detection method that can delineate the frontal warm and cold zones[37,38]. This approach enables quantitative assessment of both frontal biological hotspot effects and thermal barrier effects, and its key strength lies in its ability to isolate mesoscale frontal influences by controlling for large-scale background spatial variability through statistical indices (FRAD and $RD_{FPA}$) and a random background distribution generated by a bootstrap method[46]. Our results show that traditional statistical models fail to detect significant frontal effects in some cases, such as troller-targeted Chinook/Coho salmon (fishery 07; Fig. 4c) in the California upwelling system and albacore tuna (fishery 18) in the eastern Atlantic, as reported in previous studies[28,29], whereas our analysis captures statistically significant ($p < 0.05$) hotspot effects, with fishing activity differing by 10–20% between frontal and non-frontal zones (Fig. 4d). This discrepancy likely arises because front-induced mesoscale signals are masked by broader spatial variability in fishery distributions when using traditional statistical models[25], potentially limiting their explanatory and predictive power in fishery distribution modelling[26,47]. Beyond fisheries, the analysis framework for frontal hotspots and barrier effects holds broad potential for application to front-related processes in global and regional oceans, including impacts on marine organisms, floating pollution, and biogeochemical cycles[15,48,49], offering a tool to further understand the role of frontal dynamics in driving material circulation and energy flow within marine ecosystems[5].

It has been widely proposed that front-driven enhanced nutrient supply and primary production promote fish aggregation and consequently intensify fishing activity[9,12,21]. However, as key zones of water mass convergence with sharp thermal boundaries, frontal barrier effects—particularly at finer scales—have often been overlooked and rarely quantified due to the lack of appropriate statistical methods. Our results provide direct and quantitative evidence that ocean fronts do not universally support the aggregation of fish and fishing activities, but instead that positive hotspot effects emerge only when aggregations on either the warm or the cold side of fronts offset avoidance on the other (Fig. 5). When contrasting effects in warm and cold zones are averaged, as in previous studies that overlooked barrier effects[9,12], their neutralization results in an average global underestimation of frontal fishery impacts by 55–75%. This result provides a mechanistic understanding of why some recent studies have found weak or inconsistent links between ocean fronts and the distribution of fisheries and top predators[17,27–29]. Earlier studies suggested no clear link between 18) albacore tuna fisheries and fronts in the eastern Atlantic, whereas our results show a strong barrier effect, with fishing activity over 50% higher in warm zones than in cold zones (Fig. 4c)[29,50]. A similar pattern is also seen in Chinook and Coho salmon troll fisheries (fishery 07) in the California Current region[28]. These findings challenge prevailing assumptions about frontal hotspot effects and underscore

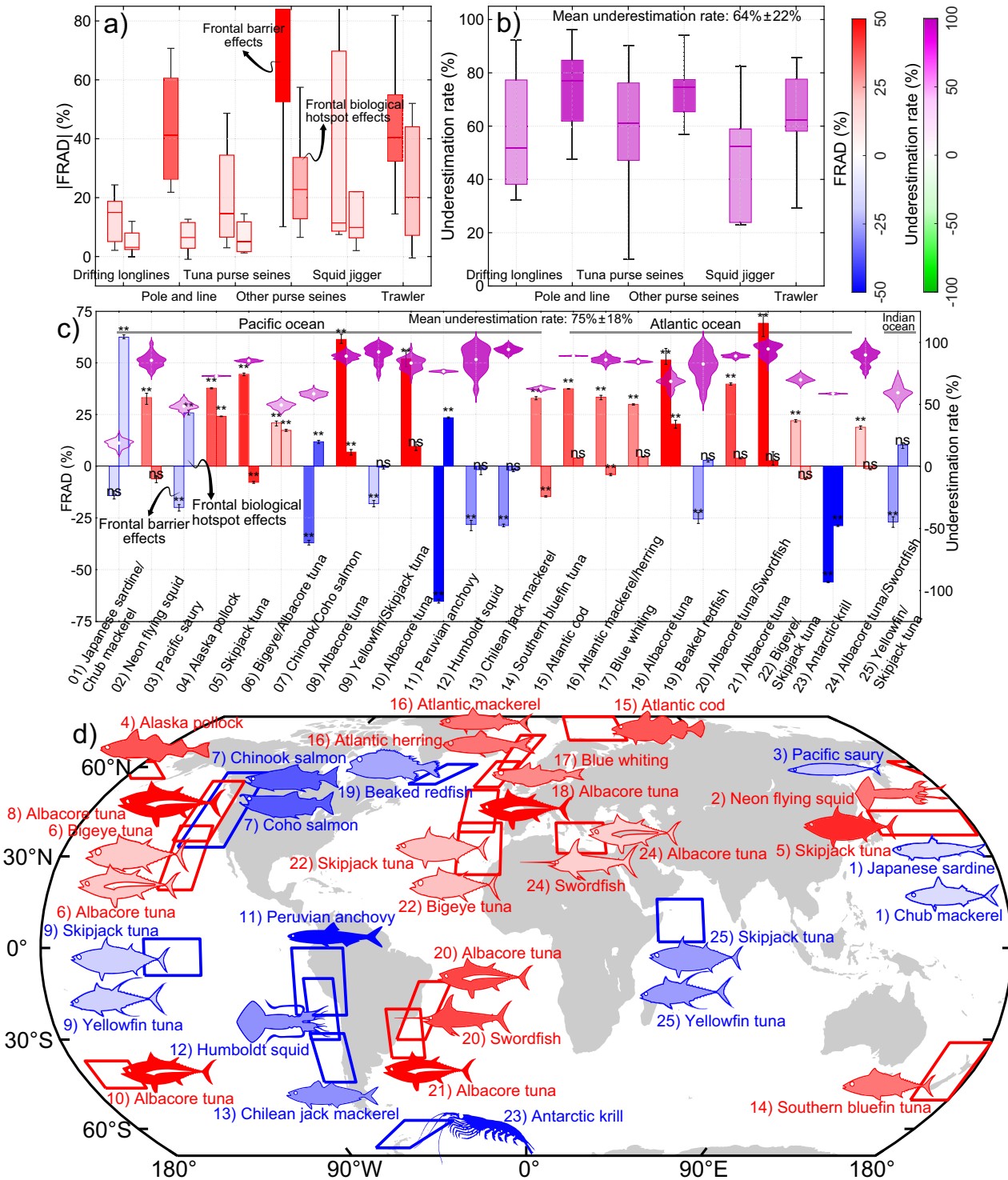

**Fig. 4 | Frontal biological hotspot and barrier effects on fisheries across different regions and stocks. a** Absolute differences in fishing effort between frontal and non-frontal zones (right, hotspot effects) and between warm and cold zones (left, barrier effects) across 11 regions and 6 fishing gears, based on the fishing effort relative anomaly difference (FRAD) results. Boxplots show the non-outlier minimum/maximum and the 25th, 50th, and 75th percentiles. **b** Underestimation rate of frontal fishery effects when barrier effects are ignored across 11 regions and 6 fishing gears, based on FRAD results. Boxplots show the non-outlier minimum/maximum and the 25th, 50th, and 75th percentiles. **c** Frontal hotspot (right bars) and barrier (left bars) effects across 25 individual fishery stocks, based on FRAD results. Positive values indicate higher fishing effort in frontal versus non-frontal zones (hotspot effects), and in warm versus cold zones (barrier effects). Error bars

represent the 25th and 75th percentiles with the central value indicating the median of 1000 simulations, and asterisks denote significance levels (*$p < 0.05$, **$p < 0.001$, ns > 0.05). Statistical significance is assessed using a one-sided paired t-test (left-tailed), without multiple comparison adjustments. The directional test is applied because the hypothesis specifically concerns underestimation rather than bidirectional differences. All exact statistics are available in the Source Data. Violin plots show the underestimation rate of frontal fishery effects when barrier effects are ignored, with white dots representing the median. **d** The geographic locations of fishing grounds for the 25 individual fishery stocks. Colors indicate the strength of frontal barrier effects based on FRAD results, with red for higher fishing effort in warm zones and blue for higher effort in cold zones. Source data are provided as a Source Data file.

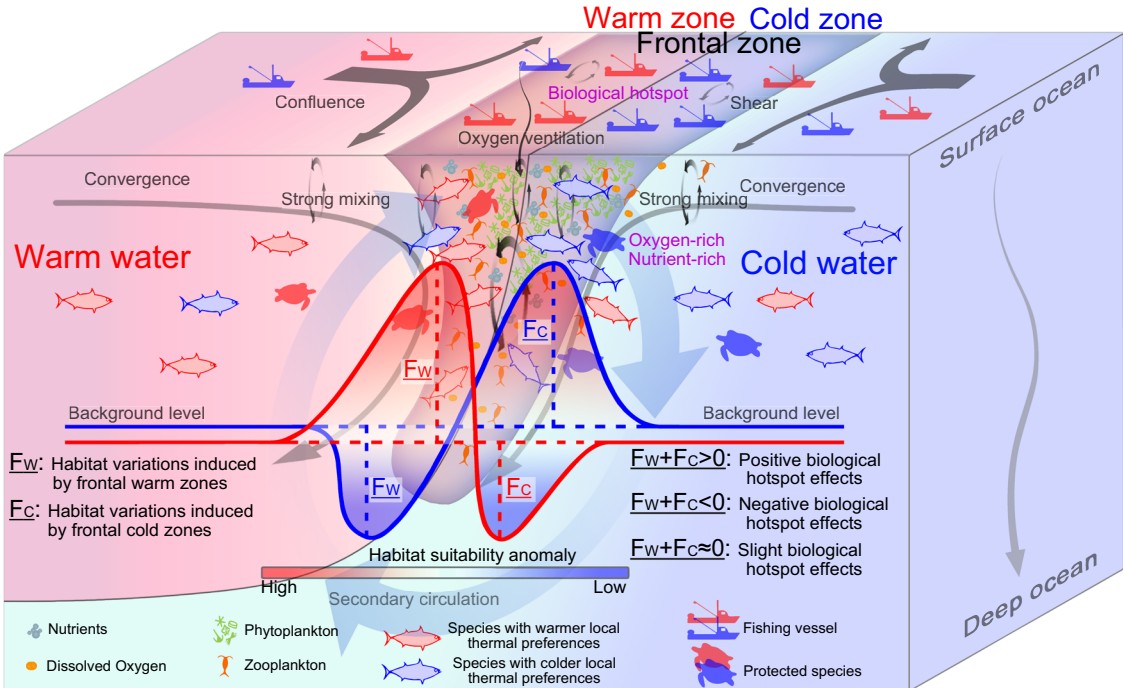

**Fig. 5 | Schematic illustrating mesoscale front-induced barrier effects and hotspot effects.** Front-driven horizontal and vertical mixing, together with secondary circulation, brings nutrients to the upper ocean and enhances oxygen ventilation, stimulating phytoplankton blooms, while convergence and confluence concentrate phytoplankton and zooplankton within frontal zones. This, in turn, attracts fish and other higher-trophic-level species. Notably, cold waters typically have higher background levels of nutrients and dissolved oxygen, as well as greater abundances of phytoplankton and zooplankton—for example, in the Kuroshio–Oyashio convergence zone investigated in this study. However, due to physiological thermal constraints, species with warmer local thermal preferences tend to aggregate in frontal warm zones, thereby attracting fishing activities to these zones despite the higher food availability in cold zones. Conversely, species with colder local thermal preferences aggregate in cold zones, similarly drawing targeted fishing effort. In this pattern, the frontal hotspot effect is positive when aggregations in either the warm or the cold zone offset avoidance on the other side; it becomes negative when these aggregations cannot offset avoidance, resulting in lower fish abundance and fishing activity in frontal zones compared to non-frontal zones. This pattern suggests a potential approach for dynamic fishery management: when target fish species and protected species prefer opposite sides of fronts, it becomes possible to balance fishing benefits with conservation goals. Source data are provided as a Source Data file.

the urgent need to reassess the role of ocean fronts in shaping fisheries.

These mesoscale front-driven boundary effects differ from the traditional understanding of large-scale frontal systems, such as the subpolar front, and the Antarctic Circumpolar Current front[51,52], and they show highly intraseasonal variability and dominate finer-scale ecosystem variations within each fishing season and area[7,8,26]. Previous studies have reported that the habitats of skipjack tuna and Pacific saury are broadly separated by the Subarctic Front[41], and Atlantic salmon migrate in warm waters separated by the Labrador Current[53]. However, our findings suggest that, at finer spatial scales, species and fisheries with distinct local thermal preferences respond strongly to highly dynamic mesoscale frontal barrier effects that overlap with hotspot effects, showing pronounced anomalies near the fronts that gradually diminish with distance (Fig. 5). Importantly, this local thermal preference differs from the traditional classification of warm- and cold-water species defined by large-scale temperature regimes. For example, Alaska pollock, cod, herring, and blue whiting are typical cold-water species but show higher fishing activity in frontal warm zones compared to cold zones (Fig. 4c)[54,55]. In contrast, Humboldt squid and yellowfin tuna, both typically warm-water species[56], exhibit greater fishing activity in frontal cold zones relative to warm zones. Meanwhile, frontal barrier effects on fisheries can even reverse with seasonal shifts in local thermal preferences, as exemplified by the contrasting north-south patterns observed in mackerel and sardine (Figs. 2h–k and 3). Such local thermal preferences may reflect shifts according to the primary physiological needs associated with seasonal migrations across life-history stages[57]. Previous studies have demonstrated that frontal barrier effects can restrict and concentrate the transport of passive tracers such as phytoplankton and zooplankton, while marine predators may at times exhibit quasi-planktonic behavior when foraging intensively in the prey-rich areas associated with fronts[12,58,59]. Our findings further extend this hypothesis by suggesting that thermal barrier effects may drive species to migrate quasi-planktonically toward either the warm or the cold side of fronts, potentially depending on their species-specific thermal ecological niches. Additionally, some predators temporarily dive into deeper, productive frontal cold zones for foraging and return to the surface warm zones to maintain body heat[33,60]. While this cross-front behavior may alter fishery catchability between warm and cold zones, these predators primarily occupy warm zones most of the time[33,60], highlighting their strong sensitivity to frontal thermal barriers.

Given their central role in key life-history processes of marine fish species[9,19,21], ocean fronts have long been used to predict fish abundance and fishery distributions over recent decades[26,61]. However, previous studies have frequently reported unexpectedly low explanatory power of fronts in species distribution models and fishery forecasts[26,47,62]; therefore, our findings call for a reassessment of the role of ocean fronts in these models, highlighting the importance of incorporating frontal barrier effects to enhance the fine-scale resolution and predictive capacity of species and fishery distributions. This has implications for improving fishers' decision-making and refining risk assessments related to fishing vessel incursions into exclusive economic zones[63]. Beyond fisheries, understanding and forecasting the fine-scale distribution of other migratory marine organisms, such as marine megafauna and protected species, could also benefit from

consideration of frontal barrier effects, especially given their differing thermal preferences and high migration capacity[17,25]. Although frontal systems have long been recognized as key feeding habitats for large marine animals, recent observations of contrasting responses among shark species[17,63], together with our findings, highlight the need to reconsider how top predators exploit these dynamic features. Previous studies have proposed dynamic ocean management strategies to mitigate bycatch of endangered species by identifying overlapping hotspots between fisheries and protected organisms[17,64], with frontal zones often prioritized due to the high co-occurrence of fish, fisheries, and vulnerable species[17,25]. Our results suggest that such management strategies can be further optimized by accounting for frontal barrier effects. Notably, they may be particularly effective when target species concentrate in either the warm or the cold side of a front, while protected species prefer the opposite, thereby reducing bycatch risks without compromising fishing opportunities (Fig. 5). A recent study reported rapidly intensifying fronts in mid- to high-latitude oceans[13], suggesting that frontal hotspot effects may become even stronger in the future and highlighting the urgency of reassessing front-induced ecosystem services under climate change.

## Methods

### Fisheries datasets and fishery-independent surveys

Given the high cost and limited spatiotemporal resolution of cruise-based surveys for highly migratory species, we compiled commercial fishing logbooks from fisheries targeting chub mackerel, Japanese sardine, neon flying squid, and Pacific saury. Data for the first three species were provided by the National Data Centre for Distant-water Fisheries of China, while data for Pacific saury were obtained directly from Chinese fishing fleets. All fishing logbooks were documented according to the standards of the North Pacific Fisheries Commission (NPFC) and were also used in previous studies[65]. These logbooks contain 109,186 daily fishing records from 2017 to 2021, documenting vessel locations and catch yields (in tons) (Fig. 1a). Chub mackerel and Japanese sardine were caught using purse seines, primarily from May to November. Neon flying squid and Pacific saury were captured using squid jiggers and stick-held dip nets, with the main fishing seasons from May to November and June to November, respectively.

To complement these data−given that Chinese vessels represent only part of the total fishing effort and do not operate within exclusive economic zones−we also used satellite-based global industrial fishing data from Global Fishing Watch (GFW, https://globalfishingwatch.org), covering the period 2012−2023. This dataset is derived from convolutional neural networks trained on automatic identification system (AIS) data, providing daily information on fishing location, effort, gear type, and Maritime Mobile Service Identity (MMSI) for each vessel[39]. We identified vessels targeting the four focal species by cross-referencing MMSI records with the NPFC vessel registry (https://www.npfc.int/), allowing us to extract an independent dataset to support our analyses[66]. Meanwhile, fishing activities involving six gear types were extracted to analyze frontal effects on fisheries across 11 globally significant regions characterized by strong and frequent oceanic fronts[43]. In addition, we extracted fishing activities targeting 25 widely studied and highly productive fishery stocks from the global industrial fishing dataset, based on gear types and fishing ground locations (Table S2, Fig. 4d), following fishery stocks in previous studies[28,55,67]. We focused exclusively on the fishing grounds of these stocks, as they exhibited stronger and more frequent frontal occurrences to better investigate frontal effects on these fisheries; in many cases, these represent only a subset of their full distribution range.

It should be noted that although variations in fishery data are often hypothesized to reflect shifts in fish distribution driven by predator−prey interactions[9,12], fishermen also use SST and SST gradient maps to locate fishing grounds, which may lead to an overestimation of frontal hotspot effects. However, there is no evidence that fishermen deliberately target the warm or cold side of mesoscale fronts, suggesting limited impacts on our results for frontal barrier effects. In addition, fishery data may also reflect catchability, which depends on the overlap between species' habitats and fishing gears[8]. Because the warm side of fronts usually shows compressed vertical habitat ranges while the cold side shows the opposite pattern due to the inclination of frontal structures (Fig. 5), such fixed catchability changes are unlikely to explain the diverse frontal fishery effects observed in this study. Although fishery data have inherent limitations, they remain one of the most suitable and widely used sources for studying mesoscale and submesoscale fish distributions across large regions, thanks to their high spatiotemporal resolution and low cost[7−9,12,26,27].

To address the limitations of fishery-dependent data, we also collected fishery-independent survey data from the Chinese research vessel Song Hang to support our results. Fishery resource surveys were conducted each year from June to July between 2021 and 2024, including midwater trawling at 163 stations and squid jigging at 67 stations. At trawling stations, midwater trawling was carried out for approximately 2−3 h at a towing speed of 4−5 knots, whereas at squid-jigging stations, squid jigging was conducted for about 5 h. The abundance index for each species was estimated as catch per unit effort (CPUE): for chub mackerel and Japanese sardine, CPUE was calculated by dividing the total catch by the volume of seawater swept by the trawl ($kg/m^3$); for neon flying squid, it was calculated as total catch per fishing hour (kg/h). More details on the cruise surveys and station distribution can be found in our previous study[68]. It should be noted that Pacific saury could not be captured during these surveys due to their distinct migratory routes[26]. Surveyed CPUEs of the three available species in warm and cold zones were paired for comparison by selecting data from adjacent frontal zones and their nearby survey stations, in order to minimize the influence of background spatial distribution and the high mobility of these species.

### Environmental datasets

To identify ocean fronts and develop thermal habitat models, we used a multi-satellite, cloud-free daily SST dataset provided by the European Space Agency (ESA, https://doi.org/10.5285/4a9654136a7148e39b7feb56f8bb02d2). This dataset, spanning 1982−2023 with a spatial resolution of 0.05° × 0.05°, integrates measurements from over 20 infrared and two microwave radiometers[69]. To assess front-induced environmental variability, we also incorporated satellite-derived sea surface chlorophyll-a concentrations and in situ subsurface profiles from Argo and BGC-Argo floats. Daily Level 4 chlorophyll-a data (2010−2023) were obtained from the Copernicus Marine Service (https://doi.org/10.48670/moi-00281), which integrates observations from most available ocean color sensors and provides a product at 0.04° × 0.04° spatial resolution. Subsurface temperature and dissolved oxygen profiles from Argo/BGC-Argo floats were sourced from the French Argo Data Centre (ftp://ftp.ifremer.fr/ifremer/argo). All profiles were quality-controlled, interpolated to 0−2000 m at 5 m vertical intervals, and smoothed using a 25 m running median filter followed by a mean filter to eliminate noise and outliers[7]. Although front-induced secondary circulation typically generates upwelling in the warm zones and downwelling in the cold zones[15], species responses to these physical processes can generally be attributed to the associated changes in environmental conditions (e.g., temperature, food availability, and dissolved oxygen levels)[7,8,46]. Therefore, this study focuses on analyzing these front-induced environmental changes rather than the physical processes themselves.

### Random simulations and significance evaluation

To isolate mesoscale front-induced fishery effects, it is necessary to remove the influence of large-scale fishery distribution patterns (i.e.,

the fishery background field). Using the original fishery dataset as a baseline, we generated 1000 simulated datasets by redistributing each real fishing record uniformly at random within a 2° × 2° window. These simulations preserve the large-scale spatial structure of fishing catches/effort and thus serve as a background reference for evaluating mesoscale variations, a method commonly applied in previous studies of mesoscale fishery distributions[7,46]. We applied our analytical framework to the observed fishery data and 1000 randomized datasets to evaluate frontal hotspot and barrier effects. Statistical significance was determined by comparing observed effects against the results of 1000 randomized simulations, with values exceeding the 95th percentile or falling below the 5th percentile considered significant at $p < 0.05$[46]. We did not use a Gaussian kernel density distribution to generate random datasets, as it may introduce bias compared with uniform randomization and does not fully remove mesoscale or submesoscale variations[27]. Meanwhile, the uniformly randomized distribution within a 2° range was designed to better capture spatial heterogeneity in frontal effects and in large-scale background distributions[46]. This differs from previous studies, which randomized organism distributions across the entire study region to facilitate region-integrated comparisons of frontal effects[27].

## Analytical framework of frontal fishery effects

Histogram-based methods have long served as the primary tool for investigating mesoscale frontal ecological and fishery effects worldwide[23], and recent advancements have addressed some limitations of earlier algorithms[37,38]. The improved algorithm enables simultaneous identification of mesoscale frontal zones and their associated warm and cold zones, offering a unified framework to assess both hotspot and barrier effects. The enhanced algorithm identifies boundaries between distinct water masses as fronts through six key steps: data preprocessing, histogram analysis, cohesion testing, frontal pixel identification, multi-window integration, and front pruning[37]. Frontal zones are then defined as the high-SST-gradient areas surrounding each front, comprising all pixels where the logarithmic SST gradient magnitude exceeds half of that at the nearest frontal pixel[38]. Within these zones, pixels warmer or colder than the nearest frontal pixel are classified as warm or cold zones, respectively. It should be noted that our frontal detection algorithm was applied using a 32-pixel box (~160 km) and should be classified as detecting mesoscale fronts, thereby excluding large-scale (>~500 km) and submesoscale (<~50 km) fronts. Meanwhile, frontal zones are commonly defined as regions of sharp environmental transitions, characterized by high gradients relative to surrounding waters[9,24]. This widely accepted but somewhat vague definition typically requires the use of subjective thresholds for detection. In our study, we adopted half of the SST gradient magnitude at frontal pixels as the threshold, following previous studies[38], but we also performed a series of sensitivity analyses using alternative values used in previous studies. Frontal zones defined with fixed widths of 40 km or 80 km, as well as those based on one-fourth or three-fourths of the SST gradient magnitude at frontal pixels, produced similar frontal barrier effects and underestimation rates to those obtained using our chosen threshold (half of the SST gradient magnitude). This indicates that our findings are robust and not sensitive to the specific threshold used for detecting frontal zones (Fig. 2b, c, Supplementary Fig. S7).

In this analytical framework, we quantified frontal hotspot effects as the difference in fishery catch/effort distribution between frontal and non-frontal zones, and barrier effects as the difference between frontal warm and cold zones. Considering the unequal spatial occurrence of frontal, non-frontal, warm, and cold zones, two independent statistical indices were used: the fishing catch (or effort) relative anomaly difference (FRAD) and the relative difference in fishing catch (or effort) per unit area (RD$_{FPA}$)[7]. The fishing catch/effort relative anomaly was defined as the relative deviation of observed total catches

or efforts from the large-scale fishery background represented by randomized simulations, and FRAD represents the difference in these anomalies between the two zones of interest[46], as calculated in Eq. (1).

$$FRAD(i,j) = \left( \frac{OF(i) - SF(i)}{SF(i)} - \frac{OF(j) - SF(j)}{SF(j)} \right) \times 100\% \qquad (1)$$

where $FRAD(i,j)$ denotes the relative anomaly difference in fishing catch or effort between zones $i$ and $j$ (frontal, warm, or cold zones), while $OF(i)$ and $SF(i)$ represent the observed and simulated fishing catch/effort in zone $i$, respectively. The term $\frac{OF(i) - SF(i)}{SF(i)}$ defines the fishing catches/effort relative anomaly in zone $i$.

Fishing catch/effort per unit area was calculated by dividing the total catch or effort within frontal, non-frontal, warm, and cold zones by the corresponding area of each zone. RD$_{FPA}$ represents the relative difference between two zones of interest[7], as defined in Eq. (2).

$$RD_{FPA}(i,j) = \frac{F(i)/A(i) - F(j)/A(j)}{F(all)/A(all)} \times 100\% \qquad (2)$$

where $RD_{FPA}(i,j)$ denotes the relative difference in fishing catch or effort per unit area between zones $i$ and $j$ (frontal, warm, or cold zones). $F(i)$ and $A(i)$ represent the total catch or effort and the total area of zone $i$, and $\frac{F(i)}{A(i)}$ defines the catch or effort per unit area in zone $i$. $\frac{F(all)}{A(all)}$ represents the value aggregated across all frontal and non-frontal zones. All RD$_{FPA}$ values were calculated at a 1° resolution to capture spatial variations, with a Gaussian filter with a 3° kernel applied to reduce noise. Additionally, we evaluated frontal barrier effects on key environmental variables at the same spatial scale using a similar method. We extracted all Argo/BGC-Argo observations located within frontal warm and cold zones for each month and calculated the averages of observed environmental variables within each 2° grid cell (due to sparser data) based on their positions. Similarly, average sea surface chlorophyll-a concentrations within the frontal warm and cold zones were calculated separately for each 1° grid on a monthly basis. Differences in epipelagic (<200 m) water temperature and dissolved oxygen concentration between warm and cold zones were derived from Argo/BGC-Argo observations, while the relative differences in sea surface chlorophyll-a concentrations were calculated by dividing the difference between satellite-derived chlorophyll-a values in frontal warm and cold zones by the mean value within each grid.

We quantified the underestimation rate of frontal fishery effects when ignoring barrier effects by calculating the percentage difference between the absolute values of hotspot and barrier effects relative to the larger of the two, as defined in Eq. (3).

$$UR = \frac{\max(|WZ|, |CZ|) - |FE|}{\max(\max(|WZ|, |CZ|), |FE|)} \times 100\% \qquad (3)$$

where $UR$ denotes the underestimation rate of frontal fishery effects when barrier effects are ignored. $WZ$ and $CZ$ represent the differences in fishery distributions between the warm and cold zones relative to the non-frontal zone, and $FE$ represents the hotspot effect, respectively, calculated as FRAD or RD$_{FPA}$ between warm, cold, or frontal zones and the non-frontal zone. The term $\max(|WZ|, |CZ|)$ refers to the greater of the absolute values of $WZ$ and $CZ$. A positive $UR$ indicates that the frontal barrier effect is underestimated relative to the hotspot effect, whereas a negative $UR$ indicates that the hotspot effect is underestimated relative to the barrier effect.

## Front-composite analysis method

To better illustrate front-induced fishery distributions, we conducted a front-composite analysis by transforming the distance from each fishing location to the nearest front into a normalized distance. This was calculated by dividing the actual distance by the frontal width,

defined as the distance from the frontal line to the boundary of frontal zones, as shown in Eq. (4).

$$ND(i) = \frac{\sqrt[2]{(x(i) - x_f(i))^2 + (y(i) - y_f(i))^2}}{\sqrt[2]{(x_b(i) - x_f(i))^2 + (y_b(i) - y_f(i))^2}} \tag{4}$$

where $ND(i)$ represents the normalized distance from fishing location $i$ to the nearest frontal pixel. $[x(i), y(i)]$ and $[x_f(i), y_f(i)]$ denote the geographic coordinates of fishing location $i$ and the nearest frontal pixel, respectively. The term $[x_b(i), y_b(i)]$ refers to the coordinates of the frontal boundary pixel nearest to frontal pixel $[x_f(i), y_f(i)]$. Unlike previous studies[47], this approach explicitly distinguishes between frontal warm and cold zones, assigning positive and negative values to the warm and cold sides, respectively. We calculated the fishing catch/effort relative anomaly across normalized distances from the front, using a bin width of 0.15. In addition, to support the fishery responses described above to their distance to fronts, we constructed a generalized additive model (GAM) using monthly mean normalized distance at a 0.25° resolution, based on the following equation: $\log(catch + 1) \sim s(normalized\ distances, by = month, k = 5) + factor(month) + \epsilon$. The analysis was performed using the R package "mgcv"[70], and the partial effects of smooth terms were extracted using the "gratia" package to evaluate fishery responses to distance from fronts[71].

## Thermal habitat suitability model

To investigate the biophysical coupling mechanisms underlying species- and season-specific frontal barrier effects, we applied a thermal habitat model to each species across fishing seasons, enabling the estimation of intra-seasonal thermal preferences and the prediction of monthly thermal habitat distributions. This ecological niche-based modeling approach, grounded in the principle of niche conservatism, has been widely used to examine the distribution of fish and other marine organisms[72,73]. SSTs at each fishing location were derived using bilinear interpolation. Total catch per 0.5 °C SST interval was aggregated and then normalized to a 0–1 scale by dividing by the maximum total catch across all intervals. The relationship between total catch and SST was modeled using a Gaussian function fitted via the least squares method, as defined in Eqs. (5–6).

$$C_n(i) = \frac{C(i)}{\max(C)} \tag{5}$$

$$THSI(i) = \frac{1}{\sigma\sqrt[2]{2\pi}} e^{-\frac{(C_n(i) - \mu)^2}{2\sigma^2}} \tag{6}$$

where $C_n(i)$ and $C(i)$ denote the normalized and original total catches for SST bin $i$, respectively, and $\max(C)$ represents the maximum total catch across all SSTs. $THSI(i)$ refers to the thermal habitat suitability index for SST bin $i$, derived from the fitted relationship between normalized catch and SST, while $\mu$ and $\sigma$ are the mean and standard deviation of the fitted Gaussian distribution, representing the thermal optimum and niche breadth, respectively. To evaluate how well SSTs in frontal warm and cold zones align with the species-specific thermal optimum, we extracted SSTs from fishing locations with the top 50% catch per unit effort (CPUE, tons/day) in each zone and compared their absolute deviations from $\mu$. Statistical significance of the differences in thermal proximity between warm and cold zones was assessed using a t-test. Meanwhile, we used the model to predict THSI values at all fishing locations with the top 50% CPUE, and compared differences between warm and cold zones across seasons and regions to underscore the central role of intra-seasonal thermal preferences in mesoscale frontal barrier effects. We focused solely on thermal habitat because front-induced barrier effects on SST exhibit the most pronounced changes and are closely associated with

front-driven variations in other environmental variables (Supplementary Fig. S4).

## Reporting summary

Further information on research design is available in the Nature Portfolio Reporting Summary linked to this article.

## Data availability

Our findings are based on open-access datasets. Our global mesoscale front dataset, together with the identified frontal warm and cold zones, is available at https://zenodo.org/records/14785322. The global fishing effort dataset is available from https://globalfishingwatch.org. The satellite-observed SST dataset can be accessed at https://doi.org/10.5285/4a9654136a7148e39b7feb56f8bb02d2. Argo and BGC-Argo profiling float data are available from https://doi.org/10.17882/42182. The satellite-derived chlorophyll-a dataset can be accessed at https://doi.org/10.48670/moi-00281. All commercial fishing record data used in this study can be obtained by applying to the National Data Centre for Distant-water Fisheries of China or by requesting them from the corresponding authors, due to fishery data privacy. Source data are provided with this paper. All maps presented in this study were generated using shoreline data extracted from the Global Self-consistent, Hierarchical, High-resolution Geography Database (GSHHG), available at https://www.ngdc.noaa.gov/mgg/shorelines/shorelines.html. Source data are provided with this paper.

## Code availability

All data analyses were conducted using MATLAB R2024a. All code for the global mesoscale front detection algorithm, along with our analysis methods for the two statistical indices used to quantify front-induced hotspot and barrier effects, is available at https://doi.org/10.5281/zenodo.17218778.

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

## Acknowledgements

This work was sponsored by the National Natural Science Foundation of China (42506078, Q.X.); the Postdoctoral Fellowship Program and China Postdoctoral Science Foundation under Grant Number BX20250007 and 2024M761926 (Q.X.); the AI Special Program of Shanghai Municipal Education Commission (A1-3405-25-000303, W.Y.); the 2024 International Cooperation Seed Funding Project for China's Ocean Decade Actions (GGZX00000, W.Y.); the Shanghai Rising-Star Cultivation Program (Sailing Program) (24YF2716700, Q.X.); the Open Funding Project of the Key Laboratory of Sustainable Exploitation of Oceanic Fisheries Resources, Ministry of Education (A1-2006-25-200202, Q.X.); the Survey, Monitoring and Assessment of Global Fishery Resources (Comprehensive scientific survey of fisheries resources at the high seas) sponsored by the Ministry of Agriculture and Rural Affairs (B.L.); the follow-up program for Professor of Special Appointment (Eastern Scholar) at Shanghai Institutions of Higher Learning (GZ2022011, B.L.); and the Japan Society for the Promotion of Science (JSPS) JP25H02072 (S.I.). We also thank Global Fishing Watch, the European Space Agency, the French Argo Data Centre, and the Copernicus Marine Service for providing publicly available datasets.

## Author contributions

Q.X. conceived and designed the study, with contributions from X.C. and W.Y.; Q.X. compiled all datasets, performed data analysis, and created the main figures, with contributions from S.I.; Q.X. wrote the original manuscript, with contributions from Z.G., S.I., H.Y., B.L., H.Z., X.C., and W.Y.; all authors contributed to writing and revising the manuscript.

## Competing interests

The authors declare no competing interests.
