## [Transparent Peer Review file · Nature Communications]

Underestimated barrier effects of ocean fronts shape global fishery distribution

Corresponding Author: Dr Qinwang Xing

Version 0:

Reviewer comments:

Reviewer #1

(Remarks to the Author)

Review of the manuscript NCOMMS-25-83163-T "Hidden barrier effects of mesoscale fronts shape global fishery distribution" by Qinwang Xing, Zihui Gao, Shin-ichi Ito, Haiqing Yu, Bilin Liu, Heng Zhang, Xinjun Chen, and Wei Yu submitted to Nature Communications

Reviewer: Igor M. Belkin

Date: 2025-11-14

Summary:

This manuscript was previously submitted to Nature Ecology & Evolution (NATECOLEVOL-25072527). I reviewed that paper. In my review I pointed out a few major flaws that made this work unpublishable. The authors were supposed to undertake a radical revision of their work to address the major flaws. Alas, the revised manuscript submitted as NCOMMS-25-83163-T falls short of expectations. The authors made numerous minor textual changes that are cosmetic in nature. No major changes have been made. All figures are the same. The manuscript's title and list of authors are the same. The abstract is almost the same. Since none of the major issues have been addressed, I have no choice but to recommend rejection.

Recommendation: Rejection

Major issues:

The entire study is based on a few major assumptions (see below) that are either incorrect or cannot be proven.

Mesoscale fronts: The emphasis on mesoscale fronts is misleading. In reality, fishermen look for high-gradient zones (fronts) because they know that fronts are associated with high productivity all the way up the food chain. Information on high-gradient zones (particularly SST fronts) is readily available from commercial companies and government agencies. Conversely, these fishermen have minimum or no information about scales of the fronts that they target. Therefore, these fishermen cannot take the front's scale into account.

Hidden barrier effects: The term "barrier" is used by the authors in a very unconventional and confusing way. When Amy Bower and Tom Rossby published their famous paper "The Gulf Stream—Barrier or Blender?" in 1985 they rigorously defined the term "barrier." The oceanographic community accepted their notion of a front as a barrier between two water bodies. The authors' usage of "barrier" is inconsistent with the standard oceanographic nomenclature. The well-known "barrier layer" has nothing to do with the "barrier" introduced by the authors. The addition of "hidden" only aggravates the terminological mess.

Barrier effect: "ectotherms may be restricted to one side of these fronts, (barrier effects), how species and fisheries respond to such barrier effects remains poorly understood." – There is a logical fallacy here. The first sentence is just an assumption. The authors hold a very simplistic view of fish ecology summed up in the optimum thermal habitat hypothesis. Fish behaviour depends on a variety of factors, and temperature is just one of them. For example, tuna forage on the cold side of fronts (by deep diving), then ascent to the surface layer and cross over the front to recuperate on its warm side (Bestley et al. (2008) wrote: "Finally, it has been suggested that when tunas forage within frontal zones, the warm side of fronts may be

used as a thermal refuge, enabling them to feed in cold waters while minimizing the cost of staying warm (Gunn & Young 1999; Kitagawa et al. 2004).” Since the restriction assumption is likely false, there is no surprise that there are no studies of a phenomenon that does not exist.

Satellite-detected warm and cold frontal zones: Such zones do not exist in the real ocean. All definitions of frontal zones (e.g., Fedorov, 1986), including the authors’ definition in this manuscript, are arbitrary and subjective.

This review is rather short because I did not want to copy and paste my previous review. The authors spent a lot of time and effort to write an extremely long rebuttal, almost certainly the longest rebuttal that I have ever seen. Their efforts are truly and greatly appreciated. Alas, all major issues pointed out in my previous review remain unresolved in the revised manuscript.

===== END of REVIEW =====

(Remarks on code availability)

Reviewer #3

(Remarks to the Author)

The authors have addressed all of my comments and concerns in a comprehensive way and I believe the clarity of the manuscript has improved. I am also appreciative of the detailed response. I have a few (minor) points I'd like to bring up and I think addressing these will streamline the paper and make it more accessible to a broader audience. I think this will be a really valuable paper once published. All the best.

First, the figures need to be more readable and a bit lighter. Specifically:

Figure 1) Red and blue for the two sides of the fronts is a bit confusing when some of the locations of the fishing sites are also in blue and red. What about using black and grey for the fronts and the colours for the different fishing targets?

Figure 4) is also a bit dense. especially c) and d). I suggest having this figure as a full page figure and having some space outside of the map and plot for the text so that it doesn't overlap with the lines of the continents too much.

Figure 5) as mentioned before by two reviewers is still very loaded. I'd suggest reducing the number of vessels and fish. It is a lot better than before, but it's still very busy.

Some text needs editing. Here is what I found, but I do recommend a careful review of the language. As a non-native speaker myself for English, I know this is extra work, but it is worthy to ensure the message of the paper gets through.

Title: Another reviewer was critical of the work 'Hidden' which I agree can be a bit confusing. What about 'Underestimated barrier effects'. Many, especially seagoing oceanographers, probably knew about these effects but I agree with the authors that they are not as well represented in the literature as the 'hotspot effects'. Levy et al., in their 2018 review paper use the language 'active front' vs 'passive front' - perhaps this is a framework that could be used in the paper here to avoid confusion?

Line 282: "driven by prey-predator interactions" - I'm not totally sure that the authors mean 'prey-predator interactions' here and not 'prey movement' or 'prey distribution'

Line 287: 'Consistently higher temperature ...' I'd specify 'compared to...' to say if the higher temperatures are higher compared to the cold side or the background or the study region.'

Line 341: I was a bit lost here when there is a reference to 57 cases. Where are these coming from? I thought the fisheries were 25. I'd also avoid the word 'cases' and I would replace it with something more specific (areas/fisheries/etc.).

Line 481: I would rephrase the comment on the quasi-planktonic movement. I don't think we know for sure why animals do that, so I would avoid saying 'to better locate and exploit enhanced food availability'. Perhaps instead we could say 'quasi-planktonic behaviour when foraging intensively in the prey-rich areas associated to fronts' or something along those lines?

Line 779: I think the first sentence of the Data availability should be streamlined as:
'Our findings are based on open-access datasets'

(Remarks on code availability)

I tried to connect to this Zenodo repository, but I got a message that authorization was required.

Version 1:

Reviewer comments:

Reviewer #1

(Remarks to the Author)

Thank you very much for addressing my concerns. Your formidable efforts are truly and greatly appreciated. Despite remaining disagreements between us, I recommend acceptance in present form.

Best regards,
Igor Belkin
2026-02-10

(Remarks on code availability)

Reviewer #3

(Remarks to the Author)

I have took part in the review of this manuscript twice and with their revisions, the authors have addressed all my concerns in a satisfactory manner. I believe that this results will be very helpful for a large community of researchers. I do not have any further concerns about this work.

(Remarks on code availability)

The code is available online on Zenodo.

These are MATLAB scripts and they are not very long and (although in a bit of minimalist way) commented.

We would like to express our sincere gratitude to the two reviewers for taking the time to review our manuscript and for providing their valuable comments. The comments from the reviewers are shown in black text, and our responses are highlighted in blue.

Responses to Reviewer #1

Q1: This manuscript was previously submitted to Nature Ecology & Evolution (NATECOLEVOL-25072527). I reviewed that paper. In my review I pointed out a few major flaws that made this work unpublishable. The authors were supposed to undertake a radical revision of their work to address the major flaws. Alas, the revised manuscript submitted as NCOMMS-25-83163-T falls short of expectations. The authors made numerous minor textual changes that are cosmetic in nature. No major changes have been made. All figures are the same. The manuscript's title and list of authors are the same. The abstract is almost the same. Since none of the major issues have been addressed, I have no choice but to recommend rejection. Recommendation: Rejection

Response: We thank you for taking the time to review our revised manuscript. We would like to clarify that we undertook a thorough and substantive revision to address all concerns raised in the previous round of review. These revisions go beyond minor textual edits and include substantial additions and modifications to the analyses and interpretation of the results. In current round, we revised the manuscript based on comments from both the previous and the current rounds. However, to keep the response at a reasonable length, we provide point-by-point replies only to the comments raised in this round. While the authorship and main figures remain unchanged, this reflects the continuity and robustness of the core findings rather than an absence of meaningful revision.

Q2: Major issues: The entire study is based on a few major assumptions (see below) that are either incorrect or cannot be proven.

Mesoscale fronts: The emphasis on mesoscale fronts is misleading. In reality, fishermen look for high-gradient zones (fronts) because they know that fronts are associated with high productivity all the way up the food chain. Information on high-gradient zones (particularly SST fronts) is readily available from commercial companies and government agencies. Conversely, these fishermen have minimum or no information about scales of the fronts that they target. Therefore, these fishermen

cannot take the front's scale into account.

Response: Thank you for your comments. Following your suggestion, we have removed the term “mesoscale” from the title and abstract. However, in the interest of transparency in peer review, as our responses may be accessible to readers, we would like to clarify that our results primarily pertain to mesoscale fronts. This is because our front detection algorithm applies a 32-pixel window (approximately 160 km) to a smoothed sea surface temperature field and identifies two distinct water masses within each 160×160 km area. As a result, the detected features are most consistent with mesoscale fronts. Our previous peer-reviewed studies on front detection algorithms have consistently adopted this interpretation (Xing et al., 2023; 2025), which has been recognized and cited by other researchers.

We agree that fishers may not explicitly distinguish the spatial scales of the fronts they target. Nevertheless, several studies have demonstrated that frontal scale is an important factor influencing the aggregation of fish and other predators. In this study, we focus on mesoscale fronts because previous work has shown that fronts with longer persistence have a greater potential to attract aggregations of fish and other predators (Scales et al., 2014; Miller et al., 2015; Morato et al., 2016; Sarma et al., 2018; Wong et al., 2026). By contrast, submesoscale fronts are typically short-lived and affect shallower water layers, and their ecological impacts are more likely to be limited to phytoplankton rather than fish and other predators. This means that not all ocean fronts are associated with high productivity, as frontal impacts depend on persistence, spatial scale, background environmental conditions, and other factors (Sarma et al., 2018; Hirzel et al., 2023). This variability may not be beneficial for fishers who do not consider the scale of fronts, which may partly explain why fishing outcomes do not always match expectations. Our earlier work further demonstrated that fish habitat models improve when submesoscale fronts are excluded and greater emphasis is placed on mesoscale fronts (Xing et al., 2022), highlighting the dominant role of mesoscale rather than submesoscale fronts in shaping fish distributions. We therefore believe that clarifying the spatial scale of the fronts considered is important for interpreting our results and for their practical relevance. Focusing on mesoscale fronts may provide greater benefits for fisheries than approaches that do not explicitly account for frontal scale, which are commonly applied at present as noted in your comments. This, in turn, highlights the potential of mesoscale fronts to better guide future fishery operations.

- Miller, P. I., Scales, K. L., Ingram, S. N., Southall, E. J., & Sims, D. W. (2015). Basking sharks and oceanographic fronts: quantifying associations in the north-east Atlantic. *Functional Ecology*, 29(8), 1099-1109.
- Morato, T., Miller, P. I., Dunn, D. C., Nicol, S. J., Bowcott, J., & Halpin, P. N. (2016). A perspective on the importance of oceanic fronts in promoting aggregation of visitors to seamounts. *Fish and Fisheries*, 17(4), 1227-1233.
- Sarma, V. V. S. S., Desai, D. V., Patil, J. S., Khandeparker, L., Aparna, S. G., Shankar, D., ... & Anil, A. C. (2018). Ecosystem response in temperature fronts in the northeastern Arabian Sea. *Progress in Oceanography*, 165, 317-331.
- Scales, K. L., Miller, P. I., Embling, C. B., Ingram, S. N., Pirodda, E., & Votier, S. C. (2014). Mesoscale fronts as foraging habitats: composite front mapping reveals oceanographic drivers of habitat use for a pelagic seabird. *Journal of the Royal Society Interface*, 11(100), 20140679.
- Wong, H. F., Schoeman, D., Miller, P. I., Bentley, L., Halpin, L., Fischer, J. H., ... & Scales, K. L. (2026). Mesoscale ocean dynamics structure fisheries interaction risk for an endangered seabird. *Biological Conservation*, 313, 111574.
- Xing, Q., Yu, H., Liu, Y., Li, J., Tian, Y., Bakun, A., ... & Li, W. (2022). Application of a fish habitat model considering mesoscale oceanographic features in evaluating climatic impact on distribution and abundance of Pacific saury (*Cololabis saira*). *Progress in Oceanography*, 201, 102743.
- Xing, Q., Yu, H., Wang, H., & Ito, S. I. (2023). An improved algorithm for detecting mesoscale ocean fronts from satellite observations: Detailed mapping of persistent fronts around the China Seas and their long-term trends. *Remote Sensing of Environment*, 294, 113627.
- Xing, Q., Yu, H., Yu, W., Chen, X., & Wang, H. (2025). A global daily mesoscale front dataset from satellite observations: in situ validation and cross-dataset comparison. *Earth System Science Data*, 17(6), 2831-2848.
- Hirzel, A. J., Alatalo, P., Oliver, H., Petitpas, C. M., Turner, J. T., Zhang, W. G., & McGillicuddy Jr, D. J. (2023). High resolution analysis of plankton distributions at the middle atlantic bight shelf-break front. *Continental Shelf Research*, 267, 105113.

Q3: Hidden barrier effects: The term “barrier” is used by the authors in a very unconventional and confusing way. When Amy Bower and Tom Rossby published their famous paper “The Gulf Stream—Barrier or Blender?” in 1985 they rigorously defined the term “barrier.” The oceanographic community accepted their notion of a front as a barrier between two water bodies. The authors’ usage of “barrier” is inconsistent with the standard oceanographic nomenclature. The well-known “barrier layer” has nothing to do with the “barrier” introduced by the authors. The addition of “hidden” only aggravates the terminological mess.

Response: Thank you for your comments and for recommending the paper. We have revised “Hidden barrier effects” to “Underestimated barrier effects” based on Reviewer #3’s suggestion. In addition, front-related studies have frequently used the

term “barrier” to describe the role of fronts in nutrient dynamics, sediment transport, heat exchange, and plankton boundaries, highlighting the significant differences between the two sides of fronts (Galarza et al., 2009; Chapman et al., 2020; Zhou et al., 2024; Yang et al., 2024; Liu et al., 2025). The paper you mentioned also adopts the term “barrier” to describe fronts as dynamical barriers that limit cross-front exchanges and maintain significant differences in water properties between the two sides of fronts. Although frontal barrier effects have rarely been discussed in the context of marine organisms and fisheries (Rintz et al., 2025), we aim to introduce the concept of barrier effects from other fields into research on marine organisms and fisheries in order to highlight significant differences in organisms and fisheries between the two sides of ocean fronts. Therefore, our use of the concept of frontal barrier effects on fisheries is essentially consistent with its use in previous front-related studies in other fields. We believe that introducing a new or alternative term could increase the complexity of understanding ocean fronts for researchers in other fields. Your recent book, *Chemical Oceanography of Frontal Zones*, also documents and supports the use of the term “barrier” in describing frontal effects on marine nutrient and chemical dynamics. Meanwhile, the term “barrier layer” has little relevance to ocean fronts, and therefore we believe that the frontal barrier effects on fisheries discussed here are unlikely to be confused with the concept of a “barrier layer,” especially given that the term “barrier” has been widely used in the front-related literature.

- Chapman, C. C., Lea, M. A., Meyer, A., Sallée, J. B., & Hindell, M. (2020). Defining Southern Ocean fronts and their influence on biological and physical processes in a changing climate. *Nature Climate Change*, 10(3), 209-219.
- Galarza, J. A., Carreras-Carbonell, J., Macpherson, E., Pascual, M., Roques, S., Turner, G. F., & Rico, C. (2009). The influence of oceanographic fronts and early-life-history traits on connectivity among littoral fish species. *Proceedings of the National Academy of Sciences*, 106(5), 1473-1478.
- Liu, Y., Yang, G., Fu, X., Li, G., Nie, L., Cui, B., ... & Wang, L. (2025). A warmer climate and an increased Yellow River runoff weakened the barrier effect of ocean fronts on the transport of terrestrial suspended sediment and nutrients in the Chinese Bohai Sea. *Journal of Marine Systems*, 104125.
- Rintz, C. L., Koubbi, P., Ramiro-Sánchez, B., Azarian, C., Caccavo, J. A., Cotté, C., ... & Leroy, B. (2025). Biogeographical Regions and Climate Change: Lanternfishes Shed Light on the Role of Climatic Barriers in the Southern Ocean. *Global Change Biology*, 31(6), e70256.
- Yang, Y., Ju, Y., Gao, Y., Zhang, C., & Lam, K. M. (2024). Remote sensing insights into ocean fronts: a literature review. *Intelligent Marine Technology and Systems*, 2(1), 10.

Zhou, X., Zhang, S., Liu, S., Chen, C., Lao, Q., & Chen, F. (2024). Thermal fronts in coastal waters regulate phytoplankton blooms via acting as barriers: A case study from western Guangdong, China. *Journal of Hydrology*, 636, 131350.

Q4: Barrier effect: “ectotherms may be restricted to one side of these fronts, (barrier effects), how species and fisheries respond to such barrier effects remains poorly understood.” – There is a logical fallacy here. The first sentence is just an assumption. The authors hold a very simplistic view of fish ecology summed up in the optimum thermal habitat hypothesis. Fish behaviour depends on a variety of factors, and temperature is just one of them. For example, tuna forage on the cold side of fronts (by deep diving), then ascent to the surface layer and cross over the front to recuperate on its warm side (Bestley et al. (2008) wrote: “Finally, it has been suggested that when tunas forage within frontal zones, the warm side of fronts may be used as a thermal refuge, enabling them to feed in cold waters while minimizing the cost of staying warm (Gunn & Young 1999; Kitagawa et al. 2004).” Since the restriction assumption is likely false, there is no surprise that there are no studies of a phenomenon that does not exist.

Response: Thank you for your comments. We have revised this sentence to avoid ambiguity, as follows: *“fronts also exhibit pronounced environmental differences between their two sides (barrier effects), and how species and fisheries respond to these effects remains poorly understood”*.

(1) We agree with your view that the restriction of ectotherms to one side of a front can be regarded as “just an assumption” (Rintz et al., 2025). Our study aims to address this gap and represents a key source of novelty by providing clear and consistent evidence that fishery distributions differ significantly between frontal warm and cold zones, based on multiple independent datasets and analytical approaches. These include: (1) two additional statistical indices with formal significance testing (Fig. 2); (2) analyses of fisheries responses as a function of distance from fronts (Fig. 2h–k); (3) partial effects from generalized additive models (Fig. S2); (4) a global fishing activity dataset derived from deep learning methods (Fig. S3a–d); and (5) four years of fishery-independent survey data (Fig. S3e). Together, these data- and method-independent analyses yield consistent results supporting the existence of frontal barrier effects on fisheries and species distributions.

(2) Meanwhile, we agree with your view that fish behavior depends on a variety of factors. As reported in previous studies, species aggregate around marine

dynamic processes, mainly due to changes in environmental factors (such as temperature, food availability, and dissolved oxygen levels) driven by these physical processes (Arostegui et al., 2022; Xing et al., 2023), which is a widely accepted hypothesis in fishery oceanography and marine ecology. However, we have examined front-induced changes in chlorophyll (a proxy for food availability) and dissolved oxygen levels, and found that these changes cannot explain the species- and season-specific responses of fisheries to fronts observed in our study, suggesting that these factors may not be the primary drivers. Meanwhile, our thermal ecological niche models produce simulations consistent with the observed species- and season-specific responses to frontal barrier effects, suggesting the local thermal preference hypothesis can explain this phenomenon well (Fig. 3). These thermal niche-based approaches have been widely applied in marine ecology and fisheries, based on the theory of ecological niche conservatism (Cheung and Pauly, 2013; Morell et al., 2024). Such ecological niche methods also form the ecological foundation for fishery prediction and species distribution models. These clarifications were incorporated into the revised manuscript in the previous round of review. Meanwhile, although our results indicate that thermal preference is a key factor underlying the observed phenomenon based on theoretical analyses and mechanistic models, we do not exclude the possibility that other, less commonly considered or as yet unrecognized, factors may also contribute.

(3) Regarding the example you mentioned about temporary cross-front behavior in tunas, we added some discussions in the revised manuscript (Lines 489–494): *Additionally, some predators temporarily dive into deeper, productive frontal cold zones for foraging and return to the surface warm zones to maintain body heat^{33,60}. While this cross-front behavior may alter fishery catchability between warm and cold zones, these predators primarily occupy warm zones most of the time^{33,60}, highlighting their strong sensitivity to frontal thermal barriers.*

Additionally, we would like to emphasize that our findings, similar to many other studies on marine predators responding to ocean dynamics and environmental variability (Queiroz et al., 2016; Scales et al., 2018; Arostegui et al., 2022; Xing et al., 2023; Lh eriau-Nice et al., 2025), should be interpreted as long-term, time-averaged patterns that reflect general rules emerging from complex and often chaotic species and fishery behaviors influenced by multiple stochastic processes. It is important not to conflate the key spatial and temporal scales of individual behaviors with those of our study on species distributions. While tunas may exhibit complex fine-scale behaviors,

such as temporary cross-front diving, their distribution probabilities over longer timescales generally show significant differences between the warm and cold sides of fronts. Indeed, Kitagawa et al. (2004) used the term “thermal barrier effect”. Although frontal barrier effects were not explicitly highlighted in the studies you cited, those results clearly indicate that tunas spend most of their time on the warmer side of fronts, with only transient excursions into colder waters (Kitagawa et al., 2004; Snyder et al., 2017). Scale is a fundamental concept in ecology, and as articulated by the renowned ecologist Levin (1992), ecological processes typically operate on specific spatiotemporal scales, with their effects evident primarily at those corresponding scales. Consequently, the fine-scale cross-front movements you mentioned may occur on shorter time scales. However, when considering fishing-hour (fishing efficiency) probabilities, the resulting fishing grounds may be located on one side of the front. Therefore, while our methods may not resolve fine-scale cross-front movements, our results reflect fish distributions at the spatiotemporal scale relevant to fisheries. Our results provide broader insight by suggesting that thermal preferences regulate species distributions between the warm and cold sides of fronts, thereby improving our understanding of their distribution patterns. We would expect that tunas with colder local thermal preferences, if present, might exhibit different cross-front movements or may not show this behavior.

- Arostegui, M. C., Gaube, P., Woodworth-Jefcoats, P. A., Kobayashi, D. R., & Braun, C. D. (2022). Anticyclonic eddies aggregate pelagic predators in a subtropical gyre. *Nature*, 609(7927), 535-540.
- Cheung, W. W., Watson, R., & Pauly, D. (2013). Signature of ocean warming in global fisheries catch. *Nature*, 497(7449), 365-368.
- Kitagawa, T., Kimura, S., Nakata, H., & Yamada, H. (2004). Diving behavior of immature, feeding Pacific bluefin tuna (*Thunnus thynnus orientalis*) in relation to season and area: the East China Sea and the Kuroshio–Oyashio transition region. *Fisheries Oceanography*, 13(3), 161-180.
- Levin, S. A. (1992). The problem of pattern and scale in ecology: the Robert H. MacArthur award lecture. *Ecology*, 73(6), 1943-1967.
- Lhériau-Nice, A., Cook, D.G. and Della Penna, A. (2025), Highly mobile pelagic species co-occur with fine-scale ocean fronts. *Limnol Oceanogr*, 70: 1901-1912.
- Morell, A., Shin, Y. J., Barrier, N., Travers-Trolet, M., & Ernande, B. (2024). Realised Thermal Niches in Marine Ectotherms Are Shaped by Ontogeny and Trophic Interactions. *Ecology Letters*, 27(11), e70017.
- Queiroz, N., Humphries, N. E., Mucientes, G., Hammerschlag, N., Lima, F. P., Scales, K. L., ... & Sims, D. W. (2016). Ocean-wide tracking of pelagic sharks reveals extent of overlap with longline fishing hotspots. *Proceedings of the National Academy of Sciences*, 113(6),

1582-1587.

- Rintz, C. L., Koubbi, P., Ramiro-Sánchez, B., Azarian, C., Caccavo, J. A., Cotté, C., ... & Leroy, B. (2025). Biogeographical Regions and Climate Change: Lanternfishes Shed Light on the Role of Climatic Barriers in the Southern Ocean. *Global Change Biology*, 31(6), e70256.
- Scales, K. L., Hazen, E. L., Jacox, M. G., Castruccio, F., Maxwell, S. M., Lewison, R. L., & Bograd, S. J. (2018). Fisheries bycatch risk to marine megafauna is intensified in Lagrangian coherent structures. *Proceedings of the National Academy of Sciences*, 115(28), 7362-7367.
- Snyder, S., Franks, P. J., Talley, L. D., Xu, Y., & Kohin, S. (2017). Crossing the line: Tunas actively exploit submesoscale fronts to enhance foraging success. *Limnology and Oceanography Letters*, 2(5), 187-194.
- Xing, Q., Yu, H., Wang, H., Ito, S. I., & Chai, F. (2023). Mesoscale eddies modulate the dynamics of human fishing activities in the global midlatitude ocean. *Fish and Fisheries*, 24(4), 527-543.

Q5: Satellite-detected warm and cold frontal zones: Such zones do not exist in the real ocean. All definitions of frontal zones (e.g., Fedorov, 1986), including the authors' definition in this manuscript, are arbitrary and subjective.

Response: Thank you for your comments. This issue was addressed through additional analyses and discussion in the previous round of review, and we carefully explain our response here for clarity.

You are correct that frontal zones and their warm and cold zones are conceptual constructs; however, the two sides of ocean fronts are objectively present due to water-mass convergence. According to a widely recognized and cited definition, a frontal zone is defined as a region of strong gradients relative to surrounding waters between two water masses with distinct properties (Fedorov, 1986), although this definition is not entirely objective and requires the selection of subjective thresholds. Within such frontal zones, the warm and cold sides naturally correspond to higher- and lower-temperature water masses, respectively. Gradient-based definitions of fronts (frontal zones) have also been used in your previous studies, where the gradient threshold values are difficult to determine objectively (Belkin et al., 2023; 2024; 2025). More broadly, numerous peer-reviewed studies have similarly adopted subjective gradient thresholds to identify frontal zones or frontal lines (Oram et al., 2008; Castelao et al., 2014; Kirches et al., 2016; Wang et al., 2021; Ren et al., 2023; Xing et al., 2024; Lhériau-Nice et al., 2025). These papers are only illustrative examples; in practice, nearly all front-related studies that require the identification of fronts or frontal zones necessarily adopt gradient or other thresholds, given the inherent ambiguity in defining

fronts.

Our method simply further subdivides frontal zones, defined using commonly applied subjective gradient-threshold approaches, into warm and cold sides; it does not introduce any additional thresholds or parameters or modify the frontal zones themselves. The need to select appropriate thresholds is therefore a recognized and unavoidable aspect of frontal research rather than a methodological limitation unique to this study, particularly given that both our method and the associated frontal dataset have been published following rigorous peer review (Xing et al., 2025). In response to your comments, we have conducted a series of sensitivity analyses using alternative definitions commonly applied in the literature (e.g., different fixed-width frontal zones and threshold values). All analyses yielded consistent and robust results (Fig. S7), indicating that our findings are not sensitive to the specific thresholds used to define frontal zones. Moreover, even if this definition is questioned, the contrasting warm- and cold-side catch patterns as a function of distance from fronts provide strong support for our conclusions (Fig. 2h–k; results based on original and normalized distances are consistent). This front-distance method does not rely on gradient-based threshold definitions and avoids the use of arbitrary or subjective thresholds. Overall, these additional analyses clearly demonstrate the robustness and reliability of our methods and results.

Additionally, we emphasize that our peer-reviewed method and datasets from previous studies provide a subjective yet transparent standard for detecting frontal zones (Xing et al., 2025), acknowledging that the definition of fronts has long been recognized as ambiguous but widely accepted (Fedorov, 1986). We believe that frontal zones need such clear standards to advance frontal research, similar to mesoscale eddies (amplitude >1 cm, area >8 pixels) and marine heatwaves (SST >90th percentile, period >5 days), which have achieved significant success using subjective yet clear criteria (Chelton et al., 2011; Hobday et al., 2016). Although not perfect, our method, for the first time, enables quantitative analysis of frontal hotspot and barrier effects using statistical models with significance testing, and has the potential to be applied to all types of front-related studies. This makes our study novel and markedly different from previous studies, which primarily relied on visual, non-quantitative approaches or experiential understanding, especially for frontal barrier effects. Importantly, our method quantifies frontal hotspot and barrier effects under the premise of the defined frontal zones, producing results that are not only consistent with certain previous

observations but also provide explanations for unexpected findings in many studies that used traditional methods. These outcomes further support the robustness of our approach. These assumptions, background, and relevant discussion have been incorporated into the Methods section of the previous revision (Lines 665–679).

- Belkin, I. M., & Zang, Y. T. (2025). South China Sea SST Fronts, 2015–2022. *Remote Sensing*, 17(5), 817.
- Belkin, I. M., Lou, S. S., & Yin, W. B. (2023). The China Coastal Front from Himawari-8 AHI SST Data—Part 1: East China Sea. *Remote Sensing*, 15(8), 2123.
- Belkin, I. M., Lou, S. S., Zang, Y. T., & Yin, W. B. (2024). The China Coastal Front from Himawari-8 AHI SST Data—Part 2: South China Sea. *Remote Sensing*, 16(18), 3415.
- Castelao, R. M., & Wang, Y. (2014). Wind-driven variability in sea surface temperature front distribution in the California Current System. *Journal of Geophysical Research: Oceans*, 119(3), 1861-1875.
- Chelton, D. B., Schlax, M. G., & Samelson, R. M. (2011). Global observations of nonlinear mesoscale eddies. *Progress in oceanography*, 91(2), 167-216.
- Fedorov, K. N. (1986). *The physical nature and structure of oceanic fronts* (Vol. 333). Berlin: Springer-Verlag.
- Hobday, A. J., Alexander, L. V., Perkins, S. E., Smale, D. A., Straub, S. C., Oliver, E. C., ... & Wernberg, T. (2016). A hierarchical approach to defining marine heatwaves. *Progress in oceanography*, 141, 227-238.
- Kirches, G., Paperin, M., Klein, H., Brockmann, C., & Stelzer, K. (2016). GRADHIST—A method for detection and analysis of oceanic fronts from remote sensing data. *Remote Sensing of Environment*, 181, 264-280.
- Lhériau-Nice, A., Cook, D. G., & Della Penna, A. (2025). Highly mobile pelagic species co-occur with fine-scale ocean fronts. *Limnology and Oceanography*.
- Oram, J. J., McWilliams, J. C., & Stolzenbach, K. D. (2008). Gradient-based edge detection and feature classification of sea-surface images of the Southern California Bight. *Remote Sensing of Environment*, 112(5), 2397-2415.
- Ren, S., Zhu, X., Drevillon, M., Wang, H., Zhang, Y., Zu, Z., & Li, A. (2021). Detection of SST fronts from a high-resolution model and its preliminary results in the south China sea. *Journal of Atmospheric and Oceanic Technology*, 38(2), 387-403.
- Wang, Y., Liu, J., Liu, H., Lin, P., Yuan, Y., & Chai, F. (2021). Seasonal and interannual variability in the sea surface temperature front in the eastern Pacific Ocean. *Journal of Geophysical Research: Oceans*, 126(2), e2020JC016356.
- Xing, Q., Yu, H., Wang, H., & Yu, H. (2023). A sliding-window-threshold algorithm for identifying global mesoscale ocean fronts from satellite observations. *Progress in Oceanography*, 216, 103072.
- Xing, Q., Yu, H., Yu, W., Chen, X., & Wang, H. (2025). A global daily mesoscale front dataset from satellite observations: in situ validation and cross-dataset comparison. *Earth System Science Data*, 17(6), 2831-2848.

Q6: This review is rather short because I did not want to copy and paste my previous

review. The authors spent a lot of time and effort to write an extremely long rebuttal, almost certainly the longest rebuttal that I have ever seen. Their efforts are truly and greatly appreciated. Alas, all major issues pointed out in my previous review remain unresolved in the revised manuscript.

Response: Thank you for your comments and for acknowledging the substantial time and effort in preparing our detailed rebuttal. We have tried to address your concerns as sincerely as possible and have done our best to present scientifically useful findings accurately. We sincerely appreciate your understanding.

Responses to Reviewer #3

Q1: The authors have addressed all of my comments and concerns in a comprehensive way and I believe the clarity of the manuscript has improved. I am also appreciative of the detailed response. I have a few (minor) points I'd like to bring up and I think addressing these will streamline the paper and make it more accessible to a broader audience. I think this will be a really valuable paper once published. All the best.

Response: We sincerely thank you for your time and effort in reviewing our manuscript and for your positive comments. We have carefully considered your suggestions and incorporated them into the revised manuscript. Please see our detailed point-to-point responses as follows.

Q2: First, the figures need to be more readable and a bit lighter. Specifically: Figure 1) Red and blue for the two sides of the fronts is a bit confusing when some of the locations of the fishing sites are also in blue and red. What about using black and grey for the fronts and the colours for the different fishing targets?

Response: Thank you for your constructive suggestion. As warm and cool colors more effectively represent frontal warm and cold zones, the red and blue have been substantially desaturated and lightened in this revision to reduce visual dominance and avoid potential confusion (Line 182).

Q3: Figure 4) is also a bit dense. especially c) and d). I suggest having this figure as a full page figure and having some space outside of the map and plot for the text so that it doesn't overlap with the lines of the continents too much.

Response: Yes, you are right. Thank you for your constructive suggestion. We

have revised and reconstructed Figure 4 based on your recommendations (Line 401). Meanwhile, we have removed the continental boundary lines and adjusted the positions of each inset, which has made the figure clearer and easier to interpret.

Q4: Figure 5) as mentioned before by two reviewers is still very loaded. I'd suggest reducing the number of vessels and fish. It is a lot better than before, but it's still very busy.

Response: Thank you for your constructive suggestion. Following your advice, we have reduced the number of vessels and fish in the figure, which has significantly improved its clarity (Line 524).

Q5: Some text needs editing. Here is what I found, but I do recommend a careful review of the language. As a non-native speaker myself for English, I know this is extra work, but it is worthy to ensure the message of the paper gets through.

Response: Thank you for your detailed review. We have carefully checked the English in the manuscript and have additionally used a large language model to help identify grammatical errors and unclear sentences. We believe that the revised manuscript shows a significant improvement in language quality. If further improvement is required, we will consult a professional English editing service.

Q6: Title: Another reviewer was critical of the work 'Hidden' which I agree can be a bit confusing. What about 'Underestimated barrier effects'. Many, especially seagoing oceanographers, probably knew about these effects but I agree with the authors that they are not as well represented in the literature as the 'hotspot effects'. Levy et al., in their 2018 review paper use the language 'active front' vs 'passive front' - perhaps this is a framework that could be used in the paper here to avoid confusion?

Response: Thank you for your insightful suggestion. We have revised it to "Underestimated barrier effects" based on your suggestion (Line 1). In addition, the terms "active front" and "passive front" appear to refer to biological growth and current-driven passive transport for phytoplankton, which we feel could be confusing when applied to the distribution of fishes and fisheries. Therefore, we believe that "barrier effects", which are commonly used to describe the limitations of cross-front exchange and the maintenance of differences in water properties between the two sides of a front, may be more appropriate. Although frontal barrier effects have rarely been discussed

in studies of marine organisms and fisheries, we aim to introduce the concept of barrier effects from other fields into research on marine organisms and fisheries in order to highlight significant differences in organism and fisheries between the two sides of ocean fronts.

Q7: Line 282: "driven by prey-predator interactions" - I'm not totally sure that the authors mean 'prey-predator interactions' here and not 'prey movement' or 'prey distribution'

Response: Thank you for your detailed review. Your understanding is correct, and we have revised it for greater clarity as follows (Line 284): "driven by a bottom-up response to prey movement".

Q8: Line 287: 'Consistently higher temperature ...' I'd specify 'compared to...' to say if the higher temperatures are higher compared to the cold side or the background or the study region.'

Response: Thank you for your detailed review. Based on your suggestion, we have added "Compared to frontal cold zones" to improve clarity (Line 289).

Q9: Line 341: I was a bit lost here when there is a reference to 57 cases. Where are these coming from? I thought the fisheries were 25. I'd also avoid the word 'cases' and I would replace it with something more specific (areas/fisheries/etc.).

Response: Thank you for your detailed review. You are correct, and we have replaced "cases" with "fisheries" to improve clarity (Line 343).

Q10: Line 481: I would rephrase the comment on the quasi-planktonic movement. I don't think we know for sure why animals do that, so I would avoid saying 'to better locate and exploit enhanced food availability'. Perhaps instead we could say 'quasi-planktonic behaviour when foraging intensively in the prey-rich areas associated to fronts' or something along those lines?

Response: Thank you for your detailed review. Following your constructive recommendation, we have revised the sentence to: "*quasi-planktonic behavior when foraging intensively in the prey-rich areas associated with fronts*" (Lines 485).

Q11: Line 779: I think the first sentence of the Data availability should be streamlined

as: 'Our findings are based on open-access datasets'

Response: Thank you for your detailed review. We have corrected the text based on your suggestion (Lines 786).

Q12: Reviewer #3 (Remarks on code availability):

I tried to connect to this Zenodo repository, but I got a message that authorization was required.

Response: We apologize for the inconvenience caused. In the previous version, we provided and tested a private link intended to grant editors and reviewers access to our code for evaluation. We are unsure why the access was restricted for you. The repository has now been made fully public at <https://doi.org/10.5281/zenodo.17218778>, and we believe it is accessible to you.

Responses to Reviewer #1

Q1: Thank you very much for addressing my concerns. Your formidable efforts are truly and greatly appreciated. Despite remaining disagreements between us, I recommend acceptance in present form.

Response: We thank you for taking the time to review our revised manuscript and for recommending its acceptance. Please feel free to contact us if you have any further questions regarding our work.

Responses to Reviewer #3

Q1: I have took part in the review of this manuscript twice and with their revisions, the authors have addressed all my concerns in a satisfactory manner. I believe that this result will be very helpful for a large community of researchers. I do not have any further concerns about this work.

Reviewer #3 (Remarks on code availability):

The code is available online on Zenodo.

These are MATLAB scripts and they are not very long and (although in a bit of minimalist way) commented.

Response: We greatly appreciate the time and effort you devoted to reviewing our manuscript and thank you for your constructive and positive feedback. Please do not hesitate to contact us if you require any further clarification.